# Combining Behaviors with the Successor Features Keyboard

**Wilka Carvalho**[*,1]    **Andre Saraiva**[2]    **Angelos Filos**[2]
**Andrew Kyle Lampinen**[2]    **Loic Matthey**[2]    **Richard L. Lewis**[3]
**Honglak Lee**[3]    **Satinder Singh**[2,3]    **Danilo J. Rezende**[2]    **Daniel Zoran**[2]
[1]Harvard University    [2]Google DeepMind    [3]University of Michigan

## Abstract

The Option Keyboard (OK) was recently proposed as a method for transferring behavioral knowledge across tasks. OK transfers knowledge by adaptively combining subsets of known behaviors using Successor Features (SFs) and Generalized Policy Improvement (GPI). However, it relies on hand-designed state-features and task encodings which are cumbersome to design for every new environment. In this work, we propose the "Successor Features Keyboard" (SFK), which enables transfer with *discovered* state-features and task encodings. To enable discovery, we propose the "Categorical Successor Feature Approximator" (CSFA), a novel learning algorithm for estimating SFs while jointly discovering state-features and task encodings. With SFK and CSFA, we achieve the first demonstration of transfer with SFs in a challenging 3D environment where all the necessary representations are discovered. We first compare CSFA against other methods for approximating SFs and show that only CSFA discovers representations compatible with SF&GPI at this scale. We then compare SFK against transfer learning baselines and show that it transfers most quickly to long-horizon tasks.

## 1 Introduction

Consider a household robot that learns tasks for interacting with objects such as finding and moving them around. When this robot is deployed to a house and needs to perform combinations of these tasks, collecting data for reinforcement learning (RL) will be expensive. Thus, ideally this robot can effectively *transfer* its knowledge to efficiently learn these novel tasks with minimal interactions in the environment. We study this form of transfer in Deep RL.

One promising method for transfer is the Option Keyboard (OK)[1, 2], which transfers to new tasks by adaptively combining subsets of known behaviors. OK combines known behaviors by leveraging Successor Features (SFs) and Generalized Policy Improvement (GPI) [3, 4]. SFs are predictive

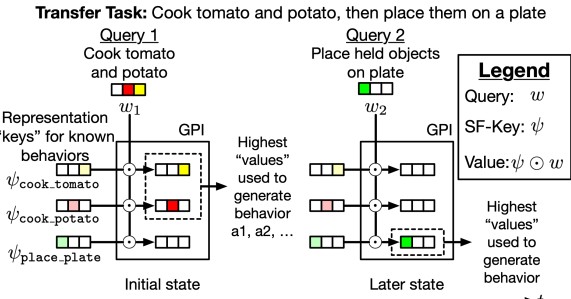

**Transfer Task:** Cook tomato and potato, then place them on a plate

Figure 1: **Diagram of transfer with the Successor Features Keyboard (SFK).** SFK uses SF-based "keys" to represent behaviors by how much of some feature (known as a "cumulant") they obtain. SFK *dynamically* selects from behaviors with GPI by generating preference "queries" over these features. The "value" of each behavior is then computed with a dot-product and the highest value is used to generate behavior. Prior work has **hand-designed** the features that define queries and SFs. Here, we study the problem of transferring with SFK while discovering all necessary representations.

---

[*]Contact author: wcarvalho@g.harvard.edu. Work done during internship.

37th Conference on Neural Information Processing Systems (NeurIPS 2023).

representations for behaviors. They represent behaviors with estimates of how much state-features (known as "cumulants") will be experienced given that behavior. GPI can be thought of as a query-key-value system that, given feature preferences, selects from behaviors that obtain those features.

While OK is a promising transfer method, it relies on hand-designed representations for the "cumulants" and task feature-preferences (i.e. task encodings). There is work that has discovered either cumulants or task encodings [3–6]. However, either (a) they only showed transfer with SF&GPI using a *fixed* (not dynamic) transfer query, (b) they only demonstrated results in simple grid-worlds where an agent combines short-horizon "goto" tasks (e.g. goto A and B), or (c) they leveraged *separate* networks for the SFs of each task-policy and for each task-encoding. This means parameter count scales with the number of tasks and limits representation re-use across tasks.

Ideally, we can transfer with a dynamic query while leveraging *discovered* representations for cumulants and feature preferences. Further, a transfer method should scale to sparse-reward long-horizon tasks such as those found in complex 3D environments. However, jointly discovering cumulants and feature preferences while estimating SFs is challenging in this setting. First, jointly learning cumulants and SFs involves estimating boot-strapped returns over a non-stationary target with high-variance and a shifting magnitude[4]. To maximize knowledge sharing across tasks, we can *share* our SF-estimator and task encoder across tasks. However, no work has yet achieved transfer results doing so.

To transfer with a dynamic query that leverages discovered representations while sharing functions across tasks, we propose two novel methods, the *Successor Features Keyboard (SFK)* and the *Categorical Successor Feature Approximator (CSFA)*. We present a high-level overview of in SFK in Figure1. SFK leverages CSFA to learn SF-estimates over discovered cumulants and task-preferences in a pretraining phase. Afterwards, in a finetuning phase, SFK learns to generate dynamic queries which are linear combinations of CSFA-discovered task-preferences. CSFA addresses challenges with estimating a non-stationary return by estimating SFs with a variant of the categorical two-hot representation introduced by MuZero [7]. We discretize the space of cumulant-return values into bins and learns a probability mass function (pmf) over them. Modelling cumulant-returns with a pmf is more robust to outliers and can better accomodate a shifting magnitude. In contrast, standard methods estimate SFs with regression of a single point-estimate [3, 8] which is succeptible to outliers and varying scales [9].

We study SFK and CSFA in Playroom [10], a challenging 3D environment with high-dimensional pixel observations and long-horizon tasks defined by sparse rewards. Prior work on transfer with SF&GPI has mainly focused on transfer in simpler 2D environments [11, 3–6]. While Borsa et al. [8] studied transfer in a 3D environment, they relied on hand-designed cumulants and task encodings [8] and only transferred to combinations of "Goto" tasks. We discover cumulants and task encodings while studying transfer to combinations of long-horizon, sparse-reward "Place near" tasks.

**Contributions**. (1) We propose the Successor Features Keyboard, a novel method that transfers with SF&GPI using a dynamic query, discovered representations, and a task encoder and SF-approximator that are shared across tasks. (2) To enable discovery when sharing a task encoder and SF-approximator cross tasks, we propose a novel learning algorithm, the Categorical Successor Feature Approximator. (3) We present the first demonstration of transfer with successor features in a complex 3D environment where all the necessary representations are discovered.

## 2   Related work on Transfer in Deep RL

Several avenues exist to transfer knowledge in Deep RL. We can transfer an agent's representations (how they represent situations), their control policy (how they act in situations), or their value function (how they evaluate situations). To transfer representations, one can learn a mapping from source domains to target domains [12], learn disentangled representations [13, 14], or learn a modular architecture [15, 16]. To transfer a control policy, some methods *distill* knowledge from a source policy to a target policy [17], others exploit *policy improvement* [18], and a third set transfer low-level policies by learning a meta-controller [19]. Finally, to transfer value functions, some approaches learn universal value functions [20] and others exploit SFs [2]. Below we review approaches most closely related to ours.

**Transferring policies**. One strategy to transfer a policy is to leverage multi-task RL (MTRL) training where you learn and transfer a goal-conditioned policy [21]. Another stratgy is to distill knowledge from one policy to another, as Distral does [22]. Distral works by learning two policies: one goal-conditioned policy and another goal-agnostic "centroid policy". The action-likelihoods of each policy are then distilled into the other by minimizing KL-divergences. Distral has strong performance in multi-task settings but it relies on the utility of a "centroid" policy for sharing knowledge across tasks. When we transfer to *longer* horizon tasks with sparse rewards, neither MTRL nor Distral may provide a policy with good jumpstart performance [23]. In this work, we study jumpstart performance and exploit successor features (SFs) with generalized policy improvement (GPI) [3].

| Method | disc. $\phi$ | disc. $w$ | query | share $w_\theta$ | share $\psi_\theta$ | 3D | transfer |
|---|---|---|---|---|---|---|---|
| Barreto et al. [3] | ✓ | ✓ | static | ✗ | ✗ | ✗ | goto-n |
| Zhu et al. [24]$^\beta$ | ✓ | ✓ | static | ✗ | ✗ | ✓ | goto-n |
| Barreto et al. [4] | ✓ | ✗ | static | ✗ | ✗ | ✓ | goto-c |
| Filos et al. [5]$^\beta$ | ✓ | ✓ | static | ✗ | ✗ | ✗ | goto-c |
| Borsa et al. [8] | ✗ | ✗ | static | ✗ | ✓ | ✓ | goto-c |
| Carvalho et al. [6] | ✓ | ✗ | static | ✗ | ✓ | ✗ | goto-c |
| Barreto et al. [1] | ✗ | ✗ | **dynamic** | ✗ | ✗ | ✗ | goto-c |
| SFK (ours) | ✓ | ✓ | **dynamic** | ✓ | ✓ | ✓ | **place-c** |

Table 1: **Related methods for transfer with SF&GPI**. SFK is the first method to transfer with a dynamic query, discover cumulants $\phi$ and task encodings $w$, while sharing a task encoder $w_\theta$ and SF approximator $\psi_\theta$ across tasks. Each of these is important to transfer with SF&GPI in a large-scale multi-task setting. Together, this enables the first SF method to transfer to combinations of long-horizon place tasks in a 3D environment with discovered $\phi$ and $w$. Note: $\beta$ refers to methods which learn from demonstration data, which we do not. goto-n is "goto new goal state", goto-c is "goto object combo", and place-c is "place object combo".

**Successor Features** are useful because they enable computing of action-values for new task encodings [3]. When combined with GPI, prior work has shown strong zero-shot or few-shot transfer to combinations of tasks. To accomplish this, GPI can evaluate known SFs with a "query" transfer task encoding. SF&GPI relies on "cumulants" (which SFs predict returns over) and task encodings that respect a dot-product relationship. Prior work has had one of three limitations. Some work has discovered representations but only shown transfer with a *static* transfer query and did not share SF-estimators or task-encoders across tasks [3, 24, 3, 5]. Other work has shared SF-estimators across tasks but exploited hand-designed task encodings with static GPI queries [8, 6]. The Option Keyboard [2] transferred using a *dynamic* query; however, they hand-designed cumulants and task encodings and didn't share functions across tasks. In this work, we present the Successor Features Keyboard, where we address all three limitations. We transfer with a dynamic query, discover cumulants and task encodings, and learn both a task-encoder and SF-estimator that are shared across tasks. Additionally, prior work has only studied transfer to combinations of short-horizon "go to" tasks whereas we include longer horizon "place" tasks and do so in a 3D environment. We summarize these differences in Table1.

## 3   Background

We study an RL agent's ability to transfer knowledge from a set of $n_\kappa$ training tasks $\mathbb{T}_{\texttt{train}} = \{\kappa_1, \ldots, \kappa_{n_\kappa}\}$ to a set of $n'_\kappa$ transfer tasks $\mathbb{T}_{\texttt{new}} = \{\kappa_1^n, \ldots, \kappa_{n'_\kappa}^n\}$. During training, tasks are sampled from distribution $p_{\texttt{train}}(\mathbb{T}_{\texttt{train}})$. At transfer, tasks are sampled from distribution $p_{\texttt{transfer}}(\mathbb{T}_{\texttt{new}})$. Each task is specified as a Partially Observable Markov Decision Process (POMDP, [25]), $\mathcal{M}_i = \langle \mathcal{S}^e, \mathcal{A}, \mathcal{X}, R, p, f_x \rangle$, where $\mathcal{S}^e$, $\mathcal{A}$ and $\mathcal{X}$ are the environment state, action, and observation spaces. Rewards are parameterized by a task description $\kappa$, i.e. $r_{t+1}^\kappa = R(s_t^e, a_t, s_{t+1}^e, \kappa)$ is the reward for transition $(s_t^e, a_t, s_{t+1}^e)$. When the agent takes action $a_t$ in state $s_t^e$, $s_{t+1}^e$ is sampled according to $p(\cdot|s_t^e, a_t)$, an observation $x_{t+1}$ is generated via $f_x(s_{t+1}^e)$, and the agent gets reward $r_{t+1}^\kappa$. We assume the agent learns a recurrent state function that maps histories to agent state representations, $s_t = s_\theta(x_t, s_{t-1}, a_{t-1})$. Given this learned state, we aim to obtain a behavior policy $\pi(s_t, \kappa)$ that maximises the expected reward when taking an action $a_t$ in state $s_t$: i.e. that maximizes

$Q_t^{\pi,\kappa} = Q^{\pi,\kappa}(s_t, a_t) = \mathbb{E}_\pi \left[ \sum_{t=0}^\infty \gamma^t r_{t+1}^\kappa \right]$. We study agents that continue learning during transfer and aim to maximize *jump-start performance* [23].

**Transfer with SF&GPI** requires two things: (1) state-features known as "cumulants" $\phi_{t+1} = \phi_\theta(s_t, a_t, s_{t+1})$, which are useful "descriptions" of a state-transition, and (2) a task encoding $w = w_\theta(\kappa)$, which define "preferences" over said transitions. Reward is then defined as $r_t^\kappa = \phi_t^\top w$ [3]. Successor Features $\psi^\pi$ are then *value functions* that describe the discounted sum of future $\phi$ that will be experienced under policy $\pi$:

$$\psi_t^\pi = \psi^\pi(s_t, a_t) = \mathbb{E}_\pi \left[ \sum_{i=0}^\infty \gamma^i \phi_{t+i+1} \right] \tag{1}$$

Given $\psi_t^\pi$, we can obtain action-values for $r_t^\kappa$ as $Q_t^{\pi,\kappa} = {\psi_t^\pi}^\top w$.

The linear decomposition of $Q^{\pi,\kappa}$ is interesting because it can be exploited to *re-evaluate* $\psi^\pi$ for new tasks with GPI. Assume we have learned SFs $\{\psi^{\pi_i}(s,a)\}_{i=1}^{n_\kappa}$ for tasks $\mathbb{T}_{\texttt{train}}$. Given a new task $\kappa'$, we can obtain a new policy $\pi(s_t; \kappa')$ with GPI in two steps: (1) re-evaluate each SF with the task's *query* encoding to obtain new Q-values (2) select an action using the highest Q-value. In summary,

$$\pi(s_t, \kappa') \in \arg\max_{a \in \mathcal{A}} \max_{i \in \{1,\dots,n_\kappa\}} \{ {\psi_t^{\pi_i}}^\top w_\theta(\kappa') \} = \arg\max_{a \in \mathcal{A}} \max_{i \in \{1,\dots,n_\kappa\}} \{ Q_t^{\pi_i,\kappa'} \}, \tag{2}$$

where $w_\theta(\kappa')$ is a **static transfer query** for transfer task $\kappa'$.

**Option Keyboard**. One benefit of equation 2 is that it enables transfer to *linear combinations* of training task encodings. However, it has two limitations. First, the feature "preferences" $w_\theta(\kappa')$ are fixed across time. When $\kappa'$ is a complex task (e.g. avoiding an object at some time-points but going towards it at others), we may want something that is state-dependent. Second, if we learn $w_\theta$ with a nonlinear function approximator such as a neural network, there is no guarentee that $w_\theta(\kappa')$ is in the span of training task encodings. The "Option Keyboard" [1, 2] can circumvent these issues by learning a transfer *policy* that maps states and tasks to a **dynamic transfer query** $g_\theta(s, \kappa')$:

$$\pi(s_t, \kappa') \in \arg\max_{a \in \mathcal{A}} \max_{i \in \{1,\dots,n_\kappa\}} \{ {\psi_t^{\pi_i}}^\top g_\theta(s_t, \kappa') \} \tag{3}$$

**Learning**. In the most general setting, we learn $\psi_\theta, \phi_\theta, w_\theta$ and $g_\theta$ from experience. Rewards $r^\kappa$ and their task encodings $w = w_\theta(\kappa)$ reference deterministic task policies $\pi_w$ that maximize them. Borsa et al. [8] showed that this allows us to parameterize an SF-approximator for a policy $\pi_w$ with an encoding of the task that defines it, i.e. we can approximate $\psi^{\pi_w}$ with $\psi_\theta(s, a, w)$. This is known as a *Universal* Successor Feature Approximator (USFA) and can be learned with TD-learning with cumulants as pseudo-rewards. To discover $\phi_\theta$ and $w_\theta$, we can match their dot-product to the experienced reward [3]. Defining $\phi_{t+1} = \phi_\theta(s_t, a_t, s_{t+1})$, we summarize this as:

$$\mathcal{L}_\psi = ||\phi_{t+1} + \gamma \psi_\theta(s_{t+1}, a_{t+1}, w) - \psi_\theta(s_t, a_t, w)||, \qquad \mathcal{L}_r = ||r^\kappa - \phi_{t+1}^\top w|| \tag{4}$$

No prior work has *jointly* learned a task encoder $w_\theta$ and USFA $\psi_\theta(s, a, w)$ while discovering cumulants $\phi_\theta$. In this work, we introduce the Successor Features Keyboard to address these limitations to enable transfer with SF&GPI and discovered representations in a large-scale 3D environment.

## 4 Method

We propose a novel method for transfer, the *Successor Features Keyboard (SFK)*, where necessary representations are discovered. To discover representations, we propose the *Categorical Successor Feature Approximator* for jointly learning SFs $\psi_\theta$, cumulants $\phi_\theta$, and task encodings $w_\theta$. The rest of this section is structured as follows. In §4.1, we describe CSFA and how to leverage it for pretraining. Finally, in §4.2, we describe how to leverage SFK for transfer. We provide background in §A.

### 4.1 Pretraining with a Categorical Successor Feature Approximator

We propose a novel learning algorithm, *Categorical Successor Feature Approximator (CSFA)*, composed of a novel architecture, shown in Figure 2, and a novel learning objective (equation 6). **Challenge**: when jointly learning a Universal SF-approximator $\psi_\theta$ and a cumulant-network $\phi_\theta$

**Categorical SF Approximator**

$\tilde{\psi}^{\pi_w}(s,a)$

Head 1 — Shared params — Head n

Structured, Categorical Network $\psi_\theta(s,a,w)$

$w$

Task Encoder

Task

State fn

Observation Encoder

$x_t$

Cumulant Network $\phi_\theta(s,a)$

$\phi_{t+1}$

$a_t$

**Transfer behavior with Successor Features Keyboard**

$\pi(s, \tau_{\texttt{new}}) = \mathrm{argmax}_a \max_{w \in \mathcal{W}_{\texttt{train}}} \{\psi_\theta(s,a,w)^\top g_\theta(s, w_{\texttt{new}})\}$

Training task behaviors we want to transfer

$w_1$ → SF network → $\psi_\theta(s,a,w_1)$ → $\odot$ → $Q^{w_1,g}$

$w_n$ → SF network → $\psi_\theta(s,a,w_n)$ → $\odot$ → $Q^{w_n,g}$

Max → Action

**GPI**

State fn

Observation Encoder

$x_t$

New State fn

$w_{\texttt{new}}$

New Task Encoder

Transfer Task

Transfer policy

$g_\theta(s, w_{\texttt{new}}) = \sum_i \alpha^i w_i$

**Legend**

Frozen parameters

Figure 2: **Left: Categorical Successor Feature Approximator (CSFA)** estimates SFs for a task encoding $w$ with a structured, categorical network. **Right: Successor Features Keyboard (SFK)** transfers by dynamically selecting combinations of known task behaviors $\mathcal{W}_{\texttt{train}} = \{w_1, \ldots, w_n\}$. SFK accomplishes this by learning a policy $g_\theta(s, w_{\texttt{new}})$ that generates linear combinations of known task encodings $\mathcal{W}_{\texttt{train}}$. SFK then leverages GPI to compute $Q$-values for known behaviors, $Q^{w_i,g} = \psi(s, a, w_i)^\top g_\theta(s, w_{\texttt{new}})$ and acts using the highest $Q$-value.

for long-horizon tasks, $\psi_\theta$ needs to fit $\phi_\theta$-generated returns that are potentially high-variance, non-stationary, and changing in magnitude [9]. CSFA addresses this challenge by modelling SFs with a probability mass function (pmf) over a discretized range of continous values. CSFA then fits this data by leveraging a categorical cross-entropy loss, enabling our estimator to give probability mass to different ranges of values. This is in contrast to prior work that models SFs with a point-estimate that is fit via regression [3, 8]. Leveraging a point-estimate can be unstable for modelling a non-stationary target with changing magnitude [9] (we show evidence in §C.1).

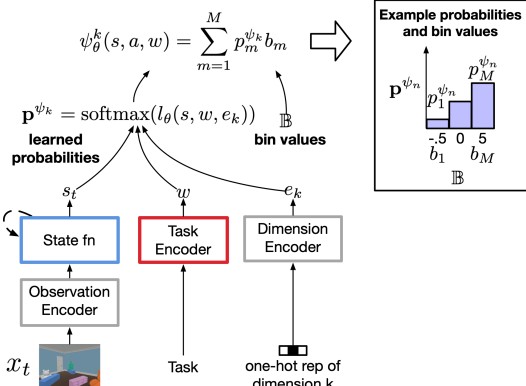

$$\psi_\theta^k(s,a,w) = \sum_{m=1}^{M} p_m^{\psi_k} b_m$$

**Example probabilities and bin values**

$\mathbf{p}^{\psi_k} = \mathrm{softmax}(l_\theta(s,w,e_k))$

**learned probabilities**

$\mathbb{B}$

**bin values**

$\mathbf{p}^{\psi_n}$, $p_1^{\psi_n}$, $p_M^{\psi_n}$

-.5 0 5
$b_1$ $b_M$

$\mathbb{B}$

$s_t$ — State fn — Observation Encoder — $x_t$

$w$ — Task Encoder — Task

$e_k$ — Dimension Encoder — one-hot rep of dimension k

Figure 3: **Diagram of how successor features can be computed with a probability mass function.**

**Architecture**. CSFA represents SFs with a pmf $(\mathbb{B}, \mathbf{p}^{\psi_k})$, where $\mathbb{B} = \{b_1, \ldots, b_M\}$ are an apriori defined set of *bin values* and $\mathbf{p}^{\psi_k}$ are probabilities for each bin value. Specifically, CSFA estimates an n-dimensional SF-vector $\psi^{\pi_w}(s,a) \in \mathbb{R}^n$ with $\psi_\theta(s,a,w)$ and represents the $k$-th SF dimension as $\psi_\theta^k(s,a,w) = \sum_{m=1}^{M} p_m^{\psi_k} b_m$. At each time-step, we update a state function $s_\theta$ with an encoding of the current observation $z_t = f^{\texttt{enc}}(x_t)$, the previous action $a_{t-1}$, and the previous state representation $s_{t-1}$, i.e. $s_t = s_\theta(z_t, a_{t-1}, s_{t-1})$. Each set of probabilities is computed as $\mathbf{p}^{\psi_k} = \mathrm{softmax}(l_\theta(s_t, w, e_k))$, where $e_k$ is an embedding for the current SF dimension. In summary,

$$\psi_\theta(s,a,w) = \{\psi_\theta^k(s,a,w)\}_{k=1}^n \quad \psi_\theta^k(s,a,w) = \sum_{m=1}^{M} p_m^{\psi_k} b_m \quad \mathbf{p}^{\psi_k} = \mathrm{softmax}(l_\theta(s,w,e_k)) \quad (5)$$

We provide a diagram of representing an SF with a pmf in Figure 3. Using a pmf allows us to re-use the same network to estimate SFs across cumulants. We hypothesize that this provides a stronger learning signal to stabilize learning across more challenging return estimates. We show evidence in Figure 6.

**Learning objective**. We learn to generate behavior by employing a variant of Q-learning. In particular, we generate Q-values using learned SFs, and use these Q-values to create targets for both estimating Q-values and for estimating SFs. For both, targets correspond to the action which

maximized future Q-estimates. We learn SFs with a categorical cross-entropy loss where we obtain targets from scalars with the $\text{twohot}(\cdot)$ operator. Intuitively, this represents a scalar with likelihoods across the two closest bins. In summary,

$$y_t^Q = r_{t+1} + \gamma \psi_{\theta^\circ}(s_{t+1}, a^*, w_{\theta^\circ}(\kappa))^\top w_{\theta^\circ}(\kappa) \qquad \mathcal{L}_Q = ||\psi_\theta(s_t, a_t, \underline{w_\theta(\kappa)})^\top \underline{w_\theta(\kappa)} - \underline{y_t^Q}||^2 \quad (6)$$

$$y_t^{\psi_k} = \phi_{\theta^\circ}^k(s_t, a_t) + \gamma \psi_{\theta^\circ}^k(s_{t+1}, a^*, w_{\theta^\circ}(\kappa)) \qquad \mathcal{L}_\psi^{\texttt{cat}} = \frac{1}{n} \sum_k \log(\mathbf{p}^{\psi_k})^\top \text{twohot}(\underline{y_t^{\psi_k}}) \qquad (7)$$

$a^* = \arg\max_a \psi(s_{t+1}, a, w)^\top w$, where $\mathbf{p}^{\psi_k} = \text{softmax}(l_\theta(s_t, \underline{w}, e_k))$, $\underline{w}$ is a stop-gradient operation on $w$. Like prior work [26], we mitigate non-stationary in the return-targets $y_t^Q$ and $y_t^{\psi_k}$ by having target parameters $\theta^\circ$ that update at a slower rate than $\theta$. The overall loss is $L = \beta_Q \mathcal{L}_Q + \beta_\psi \mathcal{L}_\psi^{\texttt{cat}} + \beta_r \mathcal{L}_r$.

**Important implementation details**. Estimating SFs while jointly learning a cumulant function and task encoder can be unstable in practice [3]. No work has jointly learned all three functions while sharing them across tasks. Here, we detail important implementation challenges that prohibited us from discovering representations that worked with SF&GPI in our large-scale setting. **D1**. We found that passing gradients to $w_\theta$ through $\psi_\theta(s, a, w)$ or through $\psi_\theta(s, a, w)^\top w$ during Q-learning lead to dimensional collapse [27] and induces a small angle between task encodings (see §C.1). We hypothesize that this makes $\psi_\theta(s, a, w)$ unstable and manifests as poor GPI performance. **D2**. When $||w||$ is large, it can make $\psi_\theta(s, a, w)$ unstable. We hypothesize that this is because it magnifies errors due to the SF-approximation error that loosen previously found bounds on GPI performance [8, 5]. We discuss this in more detail in §B. To mitigate this, we bound $w$ by enforcing it lie on a unit sphere, i.e. $w = \frac{w_\theta(\kappa)}{||w_\theta(\kappa)||}$. This makes the SF-error be consistent across tasks.

### 4.2 Transfer with the Successor Features Keyboard

The original Option Keyboard (equation 3) learned a policy that mapped states $s_t$ to queries $w$, $w' = g(s_t, \kappa^\mathrm{n})$. However, they used a hand-designed $w_\theta$ and thus hand-designed space for $w$. In our setting, we learn $w_\theta$. However, GPI performance is bound by the distance of a transfer query $w'$ to known preference vectors $w$ [3] (we discuss this in more detail in §B). Thus, we want to sample $w'$ that are not too "far" from known $w$. To accomplish this, we shift from learning a policy that samples preference vectors to a policy that samples *coefficients* $\{\alpha_i\}_{i=1}^{n_\kappa}$ for known preference vectors $\mathcal{W}_{\texttt{train}} = \{w_1, \ldots, w_{n_\kappa}\}$. The GPI query is then computed as a weighted sum. Below we describe this policy in more detail along with how to learn it.

At each time-step, the agent uses a pretrained CSFA to compute SFs $\psi^{\pi_w}(s_t, a)$ for $w_i \in \mathcal{W}_{\texttt{train}}$:

$$z_t = f_\theta(x_t) \qquad s_t = s_\theta(z_t, a_{t-1}, s_{t-1}) \qquad \{\psi^{\pi_{w_i}} = \psi_\theta(s_t, a, w_i)\}_{w_i \in \mathcal{W}_{\texttt{train}}} \qquad (8)$$

In our experiments, we freeze the observation encoder, state function, and task encoder and learn a new state function and task encoder at transfer time with parameters $\theta_\mathrm{n}$. Given a new state representation $s_t^\mathrm{n}$, we sample coefficients independently:

$$w_t' = g_{\theta_\mathrm{n}}(s_t^\mathrm{n}, w_{\theta_\mathrm{n}}(\kappa^\mathrm{n})) = \sum_{i=1}^{n_\kappa} \alpha_t^i w_i \qquad \text{where} \qquad \alpha_t^i \sim p_{\theta_\mathrm{n}}(\alpha^i | s_t^\mathrm{n}, w_{\theta_\mathrm{n}}(\kappa^\mathrm{n})) \qquad (9)$$

We find that a Bernoulli distribution performs well. We learn this $\alpha$-coefficient policy with policy gradients [28] by performing gradient ascent with gradients $A_t \nabla \log p_\theta(\alpha | s_t, w)$, where $A_t = R_t - V_{\theta_\mathrm{n}}(s_t^\mathrm{n})$ is the "advantage" of the coefficient $\alpha$ chosen at time $t$. Here, $R_t = \sum_{i=0}^\infty r_{t+i+1}$ is the experienced return and $V_{\theta_\mathrm{n}}(s_t^\mathrm{n})$ is the predicted return. Optimizing this increases the likelihood of choosing coefficients in proportion to $A_t$.

## 5 Experiments

We study transfer with sparse-reward long-horizon tasks in the complex 3D Playroom environment [10]. To transfer behavioral knowledge, we propose SFK for combining behaviors with SF&GPI, and CSFA for discovering the necessary representations. In §5.1, we study the utility of CSFA for discovering representations that are compatible with SF&GPI. In §5.2, we study the utility of SFK for transferring to sparse-reward long-horizon tasks.

**Environment setup**. We conduct our experiments in the 3D play-room environment. **Observations** are partial and egocentric pixel-based images. The agent gets no other information. **Actions**. The agent can rotate its body and look up or down. To pick up an object it must move its point of selection on the screen. When it picks up an object, it must continuously *hold it* in order to move it elsewhere. To accomplish this, the agent has 46 actions. **Training tasks**. The agent experiences $n_\kappa = 32$ training tasks $\mathbb{T}_{\texttt{train}} = \{\kappa_1, \ldots, \kappa_{n_\kappa}\}$ composed of "Find A" and "Place A near B". $|A| = 8$ and $|B| = 3$. All tasks provide a reward of 1 upon-completion. We provide more details in §F.

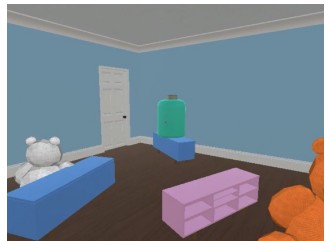

Figure 4: **Playroom**.

## 5.1 Evaluating the utility of discovered representations for SF&GPI

Our first experiments are a *sanity check* for the utility of discovered representations for SF&GPI. We train and evaluate agents on the same set of tasks. However, during evaluation the agent has access to all training tasks and must select the appropriate one with SF&GPI; given task encodings $\mathcal{W}_{\texttt{train}} = \{w_1, \ldots, w_{n_\kappa}\}$, the agent acts according to policy $\pi(s, w_i) = \arg\max_a \max_{w_k} \{\psi(s, a, w_k)^\top w_i\}$. When $w_k = w_i$, $\pi(s, w_i) = \arg\max_a Q(s, a, w_i)$. This will fail if the agent hasn't learned representations that support GPI. If an agent cannot perform GPI on training tasks, then it will probably fail with novel transfer tasks. We expand on this in §B. **Metric**. We evaluate agents with average success rate. **Challenges**. Most prior work has leveraged SF&GPI for combining "Find tasks" where an agent simply navigates to objects [4, 8, 6]. We add a significantly *longer horizon* "Place Near" task where the agent must *select* an object and *hold* it as it moves it to another object. This tests the utility of discovered representations for learning SFs that enable SF&GPI over long horizons.

**Research questions. Q1**. How does CSFA compare against baseline methods that share their SF estimatar $\psi$ across tasks while discovering $\phi$ and $w$? **Q2**. Is each piece of CSFA necessary?

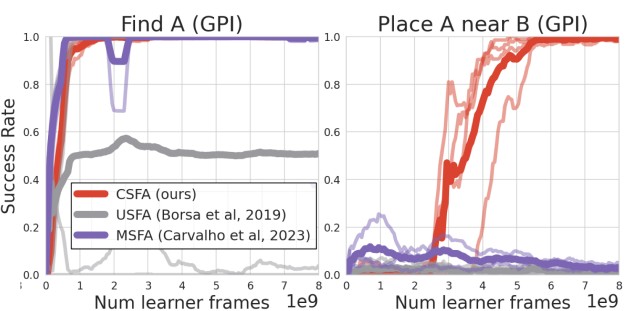

Figure 5: **CSFA discovers representations compatible with GPI across short and long task-horizons**. Both USFA and MSFA degrade in performance on Find tasks when paired with GPI. Neither are able to learn our longer horizon Place task. We hypothesize that this is because they approximate SFs with a point-estimate. (4 seeds)

**Baselines**. (1) **Universal Successor Feature Approximators (USFA)** [8] is the only method that shares an SF estimator across tasks and has shown results in a 3D environment. (2) **Modular Successor Feature Approximators (MSFA)** [6] showed that leveraging modules improved SF estimation and enabled cumulant discovery. However, they did not discover task encodings and only showed results in simple grid-worlds. **Both** baselines estimate SFs with point-estimates. Comparing to them tests (a) the utility of our categorical representation and (b) CSFA's ability to discover *both* cumulants and task encodings that enable GPI in a large-scale setting.

**CSFA discovers SFs compatible with SF&GPI while baseline methods cannot**. For fair comparison, we train each baseline with the same learning algorithm (except for SF-losses), enforce $w$ lie on a unit sphere, and stop gradients from Q-lerning. We found that large-capacity networks were needed for discovering $\phi$. In particular, we parameterized $\phi_\theta$ with a 8-layer residual network (ResNet) [29] for all methods. One important difference is that we leverage a set of ResNet modules for MSFA since Carvalho et al. [6] showed that modules facilitate cumulant discovery. Figure 5 shows that USFA and MSFA can perform GPI for Find tasks; however, neither learn place tasks in our computational budget. Given that we controlled for how $\phi$ and $w$ are learned, we hypothesize that the key limitation of USFA and MSFA is their reliance on scalar SF estimates.

**A categorical representation is necessary**. One difference between MSFA/USFA and CSFA is that CSFA shares an estimator parameters across individual SFs. Figure 6 shows that when CSFA shares an estimator but produces scalar estimates (CSFA - no categorical), GPI performance degrades

*below* MSFA/USFA. We hypothesize that a network producing point-estimates has trouble estimating returns for cumulants of varying magnitude. In §C we show evidence that CSFA has more stable SF-errors compared to scalar methods.

**Sharing our estimator across cumulants is necessary**. If we keep our categorical representation but don't share it across cumulants (CSFA - independent), GPI performance degrades on our long-horizon place near task. **Stopping gradients from Q-learing is necessary**. Interestingly, when we pass gradients to the task-encoder from Q-learning (CSFA - no stop grad), we can get perfect train performance on place tasks but highly unstable GPI performance. We found that passing gradients leads to dimensional collapse [27] (see §C). We hypothesize that this makes $\psi_\theta$ unstable. Likewise, if we don't bound tasks (CSFA - no $||w||$), we find degraded GPI performance compared to training but for even simpler tasks. While other methods may work, enforcing $w$

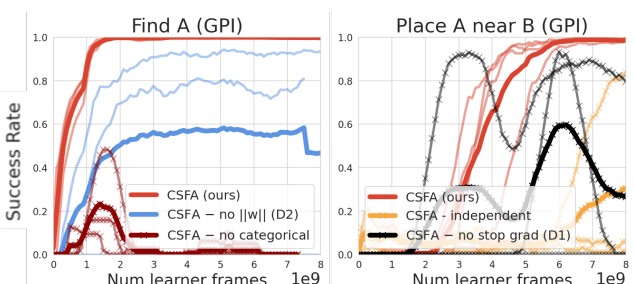

Figure 6: **CSFA Ablations. Left**. Not bounding $||w||$ (**D2**) degrades GPI performance. Interestingly, removing our categorical representation (CSFA - no categorical) gets perfect training performance but terrible GPI performance. **Right**. Passing gradients to the task-encoder from the SF-approximator leads to unstable GPI performance (**D1**) despite good training performance. If CSFA does not share its SF-estimator, it seems to learn more slowly. Thankfully, each addition is simple and together enables GPI with long-horizon place tasks. (3 seeds)

lie on a unit-sphere is a simple solution. We present full results for these plots in §D.

## 5.2 Transferring to combinations of long horizon tasks

Our second experiments test the utility of SFK for transferring to combinations of long horizon, sparse-reward tasks. **Transfer tasks** are conjunctions of known tasks. Subtasks can be completed in any order but reward is only provided at task-completion. Find tasks contribute a reward of 1 and place tasks contribute a reward of 2. We provide more details in the §F.

**Research questions**. **Q3**. How do we compare against baseline transfer methods? **Q4**. How important is it to use CSFA and to sample coeffects over known task encodings?

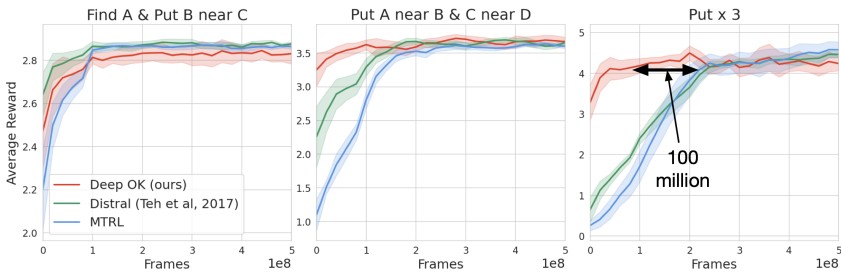

Figure 7: **SFK transfer most quickly to longer horizon tasks**. Distral and MTRL learn at about the same speed, though Distral is slightly faster. For put x 3, where the agent needs to do 3 place tasks (A near B, C near D, and E near F) SFK learns with 100+ fewer samples. (9 seeds)

**Baselines**. (1) **Multitask RL (MTRL)**. We train an Impala [30] agent on all training tasks and see how this enables faster learning for our transfer tasks. We select Impala because it is well studied in the Playroom environment [31, 32, 10]. (2) **Distral** [22] is a common transfer learning method which learns a centroid policy that is distilled to training task policies. Comparing against MTRL and Distral tests the utility of combining behaviors with SF&GPI.

**Q3: SFK has better jumpstart performance**. Figure 7 shows that all methods get similar performance by 500 million frames. However, for longer task combinations (Put 2 times or 3 times), SFK gets to similar performance with far fewer frames. When we remove our curriculum (Figure 8)

this gap further increases. No method does well for our longest task (Put x 4) which involves 8 objects. We conjecture that one challenge that methods face is holding an object over prolonged periods of time. If the agent selects the wrong action, it will drop the object it's holding. This may make it challenging when there's some noise from either (a) centroid task, as with Distral, or (b) SF&GPI as with SFK. Despite not reaching optimal performance, SFK provides a good starting point for long-horizon tasks.

**Q4: Leveraging CSFA and sampling coefficents over known task encodings is critical to jumpstart performance**. Figure 9 shows us that if we don't leverage CSFA to estimate SFs and instead use USFA, SFK fails to transfer. We hypothesize that this is because of the USFA uses point-estimates for SFs, which shows poor GPI on training tasks (see Figure 5) so its not surprising it fails on novel transfer tasks. Directly sampling from the encoding space does not transfer as quickly. Empirically, we find that learned task encodings have a high cosine similarity, indicating that they occupy a small portion of the encoding space (see §C). We hypothesize that this makes it challenging to directly sample in this embedding space to produce meaningful behavior.

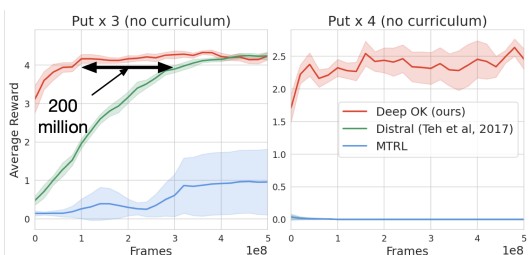

Figure 8: **SFK transfers most quickly with no curriculum of shorter tasks**. SFK and Distral reach the same performance but SFK is 200+ million frames faster. MTRL fails to transfer here. For "Put x 4", SFK is suboptimal but achieves some success. (8 seeds)

# 6 Discussion and conclusion

We have presented SFK, a novel method for transfer that adaptively combines known behaviors using SF&GPI. To discover representations that are compatible with SF&GPI, SFK estimates SFs with a novel algorithm, CSFA. CSFA constitutes both a novel architecture (of the same name) which approximates SFs with a pmf over a discretized continous values and a novel learning objective which estimates SFs for discovered cumulants with a categorical cross-entropy loss.

We first showed that CSFA is able to discover SF&GPI-compatible cumulants and task encod-

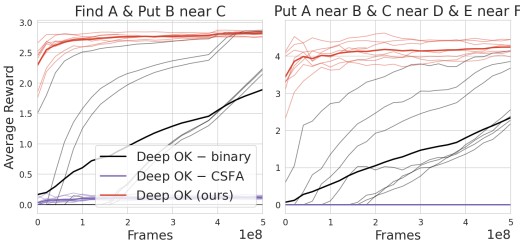

Figure 9: **SFK ablation**. SFK fails to transfer if we remove CSFA. We find that sampling directly in task-encoding space instead of sampling binary coefficents is both slower to learn and has higher performance variance. We hypothesize that this is due to concentration in the task encoding space.

ings for long-horizon sparse-reward tasks in the 3D Playroom environment (§5.1). We compared CSFA to other methods which share their approximator across tasks: (1) USFA, which showed results in a 3D environment but hand-designed representations, and (2) MSFA, which discovered cumulants but did so in gridworlds with hand-designed task encodings. We additionally compared against an ablation which removed our categorical representation and or did not share it across cumulants. Our results show that a categorical representation over discretized values can better handle estimating SF-returns when discovering cumulants and task encodings for long-horizon sparse-reward tasks.

We built on these results for our second set of experiments and showed that SFK provides strong jumpstart performance for transfer to combinations of training tasks (§5.2). We compared SFK to (1) Mulitask RL (MTRL) pretraining and finetuning, and (2) Distral, which distills knowledge back and forth between a task-specific policy and a "centroid" policy. Our results showed that, for long-horizon tasks, SFK could transfer with 100 million+ fewer samples when a curriculum was present, and 200 million+ fewer samples when no curriculum was present. We found that simply using a Bernoulli distribution for sampling task coeffcents performed well because it facilitates exploiting SF&GPI. SF&GPI enable transfer to linear combinations of task encodings. By leveraging a Bernoulli distribution, the dynamic transfer query was simply a linear combination of learned task encodings. This is precisely the type of transfer task encoding that SF&GPI has been shown to work well with.

**Limitations**. While we demonstrated discovery of cumulants and task encodings that enabled transfer with a dynamic SF&GPI query, we relied on generating queries as weighted sums of known task encodings. A more general method would directly sample queries from the task encoding space. We found that the task encodings we discovered were fairly concentrated. Future work may mitigate this with contrastive learning [33]. This respects the dot-product relationship of cumulants and task encodings while enforcing they be spread in latent space. Finally, while we demonstrated good jumpstart performance for long-horizon tasks, we did not reach optimal performance in our sample budget. Despite building on the Option Keyboard, we did not employ options as we sampled a new query at each time-step. Future work may improve performance by sampling queries more sparsely and by implementing a latent "initiation set" [34].

This paper did not study the upper limit on the number of main tasks our method can handle. However, it was a significant increase in complexity from prior work [3, 4, 8, 6, 1]. For example, if one combines 2 "go to"tasks, this leads to $C(4, 2) = 6$ combinations. We have 8 find tasks and 24 place near tasks, where $C(n, k)$ denotes $n$ choose $k$ combinations. If we combine just 2 place near tasks, this leads $C(24, 2) = 276$ possible transfer combinations. We are optimistic that one can scale the capacity of a sufficiently expressive task encoder with the number and complexity of tasks being learned.

**Conclusion**. These results present the first demontration of transfer with the Option Keyboard (SF&GPI with a dynamic query) when all representations are discovered and when the task-encoder and SF-approximator are *shared* across tasks. This may enable other methods that leverage SFs in a multi-task settings to leverage discovered representations (e.g. for exploration [35] or for multi-agent RL [5, 36]). More broadly, this work may also empower neuroscience theories that leverage SFs as cognitive theories to better incorporate discovered representations (e.g. for multitask transfer [37] or for learning cognitive maps [38]). We are hopeful that SFK and CSFA will enable SFs to be adopted more broadly with less hand-engineering.

# 7 Acknowledgements

The authors would like to thank the anonymous reviews for helpful comments in improving the paper and its accessibility. We also thank members of Google DeepMind for their helpful feedback.

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

# A Learning to estimate scalar values with distributional losses

We consider the setting where we are learning a function that maps an input $x \in \mathcal{X}$ to a scalar $y \in \mathbb{R}$.

**Representing scalars with probability mass functions (pmfs)**. One can represent any scalar $y$ as a weighted average of some of fixed bins $\mathbb{B} = \{b_1, \ldots, b_M\}$ and learnable weights $\mathbf{p} = \{p_1, \ldots, p_M\}$ as

$$y \approx \sum_{m=1}^{M} p_m b_m \tag{10}$$

where $b_m \in \mathbb{R}$ and $p_m \in \mathbb{R}^+$. $(\mathbb{B}, \mathbf{p})$ jointly define a pmf. We consider the setting where $\mathbb{B}$ is defined apriori.

**Learning pmf representations of scalars**. We can learn the weights $\mathbf{p}$ with a neural network $f_\theta$ that computes $\mathbf{p} = \mathrm{softmax}(f_\theta(x)) \in \mathbb{R}^M$ by using a negative log-likelihood objective [39]. MuZero [7] showed that one can construct an effective target from a scalar $y$ with the two-hot representation $\mathrm{twohot}(y) \in \mathbb{R}^M$. The resultant objective is then $\mathcal{L}_{\mathrm{NLL}} = \mathrm{twohot}(y)^\top \log(\mathbf{p})$. In summary,

$$\mathbf{p} = \mathrm{softmax}(f_\theta(x)) \in \mathbb{R}^M \tag{11}$$

$$\mathcal{L}_{\mathrm{NLL}} = \mathrm{twohot}(y)^\top \log(\mathbf{p}) \tag{12}$$

Prior literature has shown that leveraging this strategy for representing value-functions can improve learning (e.g. by stabilizing gradients) [39, 7].

**The two-hot encoding** $\mathrm{twohot}(y)$ is a generalization of the one-hot encoding where all elements are 0 except for the two entries closest to $y$ at positions $m$ and $m + 1$. These two entries sum up to 1, with more weight given to the entry that is closer to $y$:

$$\mathrm{twohot}(y)_i \doteq \begin{cases} |b_{m+1} - y| \, / \, |b_{m+1} - b_m| & \text{if } i = m \\ |b_m - y| \, / \, |b_{m+1} - b_m| & \text{if } i = m + 1 \\ 0 & \text{else} \end{cases} \tag{13}$$

**Representing a successor feature with a pmf**. In our setting, we approximate an n-dimensional SF-vector $\psi^{\pi_w}(s, a) \in \mathbb{R}^n$ with $\psi_\theta(s, a, w)$. We learn a different set of weights $\mathbf{p}$ for each dimension and assume all SF values lie between $b_1$ and $b_M$ of an apriori defined set of bin values $\mathbb{B} = \{b_1, \ldots, b_M\}$. We'll reference the weights for the $k$-th SF dimension as $\mathbf{p}^{\psi_k} = \{p_1^{\psi_k}, \ldots, p_M^{\psi_k}\}$. For the $k$-th SF dimension, we compute these weights as $\mathbf{p}^{\psi_k} = \mathrm{softmax}(l_\theta(s, w, e_k))$, where $s$ is a state representation, $w$ is a task representation, and $e_k$ is an embedding of that SF dimension. We can then compute the k-th SF as $\psi_\theta^k(s, a, w) = \sum_{m=1}^{M} p_m^{\psi_k} b_m$. We present a schematic of this in Figure 3.

# B Challenges of SF&GPI with learned representations

Our goal is to learn a successor feature (SF) estimator $\psi_\theta(s, a, w)$ that (a) uses the output of a cumulant function $\phi = \phi_\theta(s, a)$ as its prediction target and (b) is parameterized by the output of a task encoder $w = w_\theta(\kappa)$. In this section, we will first describe how these terms show up in the performance bounds for GPI (equation 15). Afterwards, we will describe how these terms show up in empirical challenges we faced when trying to learn these values for sparse-reward long-horizon tasks (§B.1). We will conclude with how these challenges manifest in the simplified setting of §5.1 where we "transfer" to known training tasks (§B.2).

**GPI performance bound**. Given learned $\psi_\theta, \phi_\theta$, and $w_\theta$, we want to leverage GPI to transfer to task encoding $w' = w_\theta(\kappa')$. In this setting we have learned $n$ policies $\{\pi_{w_i}\}_{i=1}^n$ for $n$ tasks $\mathcal{W} = \{w_i\}_{i=1}^n$, and can leverage GPI to transfer to $w'$ as follows

$$\pi(s, w') = \arg\max_a \max_{w \in \mathcal{W}} \{\psi(s, a, w)^\top w'\} \tag{14}$$

Define $\pi^*$ as the optimal policy for task $\kappa'$. Filos et al. [5] showed that the gap between a tasks's optimal Q-value $Q^*(s, a, w')$ and the Q-value for the GPI-policy $\tilde{Q}^\pi(s, a, w')$ can be bound by (1) the SF approximation error $\delta_\Psi = ||\psi^{\pi_w} - \tilde{\psi}^{\pi_w}||_\infty$, (2) the reward approximation error $\delta_r =$

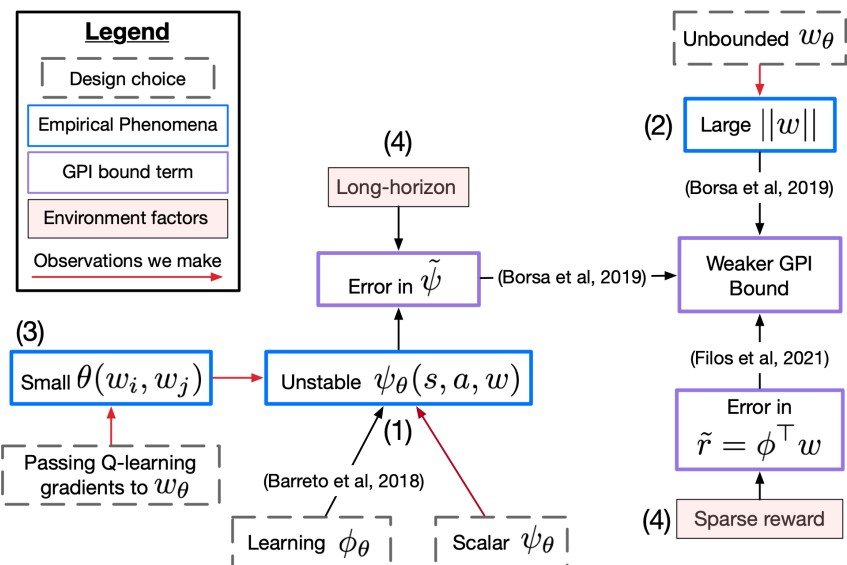

Figure 10: **Summary of GPI challenges when jointly learning SFs $\psi_\theta$, cumulants $\phi_\theta$, and task encodings $w_\theta$.** (1) Estimating SFs over non-stationary cumulants with a point-estimate can make $\psi_\theta$ unstable (see Figure 13) (2) Large task encoding magnitude $||w||$ amplifies the effect of the SF-error in the GPI bound (see equation 15) (3) Task encodings exhibit dimensional collapse when they get Q-learning gradients (see Figure 15b) (4) Sparse reward, long-horizon tasks can exacerbate errors in approximating rewards and SFs. See §B.1 for more.

$||r^{\kappa'} - \phi^\top w'||_\infty$, (3) the magnitude of the transfer encoding $||w'||$, (4) distance of $w'$ to the closest training task, $\Delta_w = \min_{j \in \{1,...,n_\kappa\}} ||w' - w_j||$. That is, $\forall s, a$

$$Q^*(s,a,w') - \tilde{Q}^\pi(s,a,w') \leq \frac{2}{1-\gamma} \left[ ||\phi||_\infty \Delta_w + ||w'|| \, \delta_\Psi + \frac{(2-\gamma)\delta_r}{(1-\gamma)} \right] \tag{15}$$

Intuitively, this says that the error on the esimated Q-values for task encoding $w'$ will grow in proportion to (1) the distance of the task encoding $w'$ to known tasks $\Delta_w$, (2) the product of the SF-approximation error and task encoding magnitude $||w'||\delta_\Psi$, and (3) the reward-approximation error $\delta_r$.

### B.1 Empirical observations

We found some design choices can weaken GPI performance and connect then to equation 15 (e.g. see Figure 6 for a summary).

**Challenge 1**. The first challenge, and the main focus of this paper, comes from estimating SFs with a scalar point-estimate while jointly discovering cumulants. Empirically, we found that cumulants changing in magnitude over the course of learning. As cumulants change in magnitude, their corresponding return can quickly change (see Figure 12). Empirically, we found that estimators that use scalar point-estimates have oscillating SF-errors $\delta_\Psi$ (see Figure 13). While they can learn training tasks, they failed to produce SF-estimates that work with GPI (see Figure 14).

**Challenge 2**. GPI performance is bounded by product of the SF-approximation error and task encoding magnitude $||w'||\delta_\Psi$. Empirically, we found that if we did not bound $||w||$, it can have a relatively large magnitude (see Figure 15a). In figure 11, we provide intuition for why this can inhibit GPI performance. We confirm this empirically in Figure 20.

**Challenge 3**. In this work, we focus on learning an SF-approximator parameterized by a task-encoder $\psi_\theta(s, a, w)$. When passing gradients to the task-encoder from Q-learning with SF-approximator, we found that this lead to dimensional collapse [27] (see Figure 15b). We conjecture that dimensional collapse can make the SF-estimator unreliable for differentiating between training tasks since $w$ is an input. This may manifest with poor GPI performance as shown in Figure 21.

## B.2 Why might SF&GPI fail for transferring to training tasks?

Our first experiments showed that jointly learned SFs $\psi_\theta$, cumulants $\phi_\theta$, and task encodings $w_\theta$ can exhibit poor GPI performance for transferring to training tasks (§5.1). Here, we provide intuition for why this can happen.

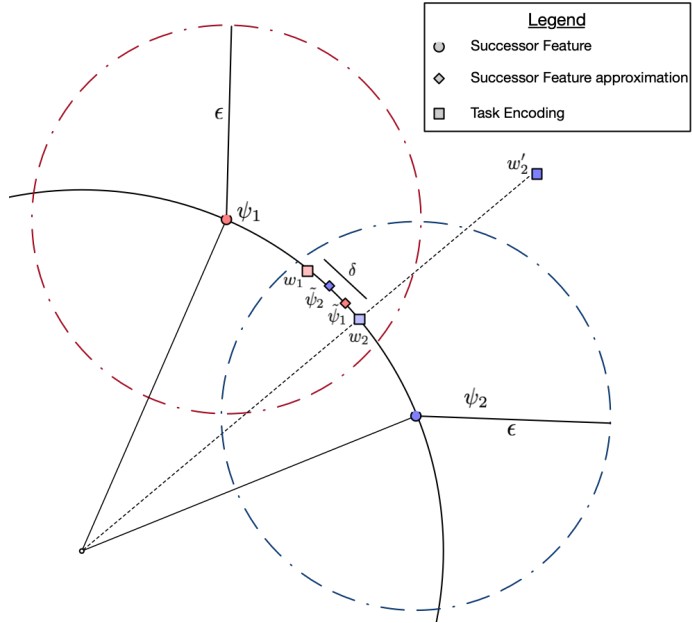

Figure 11: **Illustration of why SF&GPI can fail for transferring to training tasks**. Define $\psi_i = \psi(s, a, w_i)$, $\tilde{\psi}_i = \psi_i \pm \epsilon$. When transferring to training tasks, the correct task encoding for $\tilde{\psi}_i$ is $w_i$. Assume we are transferring to $w_2$. GPI will select the wrong task if $\tilde{\psi}_1^\top w_2 > \tilde{\psi}_2^\top w_2$. This happens due to (a) error in the SF-approximatin or (b) the magnitude of $w_2$.

When we transfer to a train task, the agent simply needs to act with the SF corresponding to the task policy, i.e. we want task selected in the inner max $w_{\texttt{sel}} = \max_w \psi(s, a, w)^\top w_i = w_i$. In this setting, the distance to the closest training task is 0, i.e. $\Delta_w = \min_{j \in \{1, \dots, n_\kappa\}} ||w' - w_j|| = 0$. This leads to the following change to equation 15.

$$Q^*(s, a, w') - \tilde{Q}^\pi(s, a, w') \leq \frac{2}{1 - \gamma} \left[ ||w'|| \, \delta_\Psi + \frac{(2 - \gamma)\delta_r}{(1 - \gamma)} \right] \tag{16}$$

This tells that, **even when we are transferring to a known task**, performance is bound by (1) the product of the SF-approximation error and task encoding magnitude $||w'||\delta_\Psi$, and (2) the reward-approximation error. The second term is intuitive—if we have not learned cumulants that predict reward, then we cannot learn SF-approximators that predict action-values for task reward. Below, we unpack the first term.

Define $\psi_i = \psi(s, a, w_i)$. Now assume that approximations for SFs are bounded by an error $\epsilon$, i.e. $\tilde{\psi}_i = \psi_i \pm \epsilon$. Now consider a simplified setting with only two tasks $w_1$ and $w_2$ and their SFs $\psi_1$ and $\psi_2$, shown in Figure 11. Consider computing GPI for $w_2$. **First**, simply due to the error $\epsilon$, we can have that $\tilde{\psi}_1^\top w_2 > \tilde{\psi}_2^\top w_2$. **Second**, if $||w||$ is large, it can dominate the dot-product and lead to $\tilde{\psi}_1^\top w_2' > \tilde{\psi}_2^\top w_2'$.

# C Additional analysis

Here we present some additional analysis. In §C.1, we show additional analysis in a simplified curriculum of only "Find" tasks. In §C.2, we show more ablations for our method.

### C.1 Analysis demonstrating observed learning challenges

We conduct additional analysis on $n_\kappa = 8$ training tasks $\mathbb{T}_{\texttt{train}} = \{\kappa_1, \ldots, \kappa_{n_\kappa}\}$ composed of "Find A" where $|A| = 8$. All tasks provide a reward of $1$ upon-completion. This is a simplified version of the experiments in §5.1 where we study some of the challenges observed during training. Specifically, we present evidence for the following:

1. The magnitude of cumulants changes during learning (see Figure 12).

2. The SF TD-error can be more unstable for methods that estimate SFs with a point-estimate (see Figure 13).

3. $||w||$ can be relatively large if we don't bound it (see Figure 15a).

4. Passing gradients to the task encoder leads to dimensional collapse (see Figure 15b)

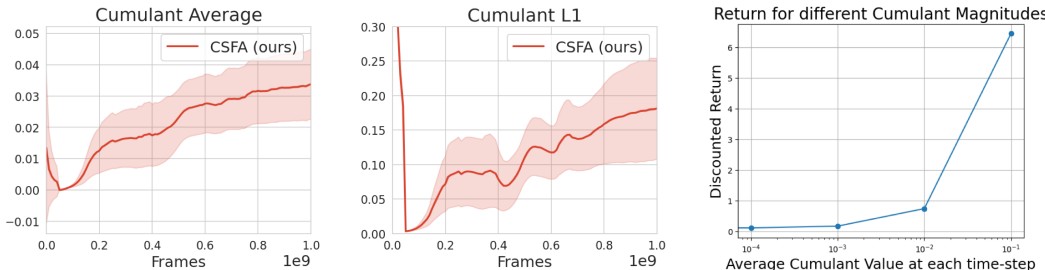

Figure 12: **Evidence that the magnitude of cumulants and cumulant-returns changes over the course of learning**. The left and middle plots show that the cumulant average and the cumulant L1 ($\bar{\phi}$ and $||\phi||_1$, respectively) both increase during training. The right-most plot shows the discounted return that would be calculated over 100 time-steps for different values of $\bar{\phi} \in \{10^{-4}, 10^{-3}, 10^{-2}, 10^{-1}\}$. We calculate return as $R_{\bar{\phi}} = \sum_{i=0}^{100} \gamma^i \bar{\phi}$, where $\gamma = .99$. We emphasize that this is not reflective of the true returns experienced by the agent but instead demonstrates how the return *magnitude* can change dramatically as the cumulant average changes. In Figure 13, we show that methods which estimate this return with a scalar point-estimate exhibit unstable learning dynamics.

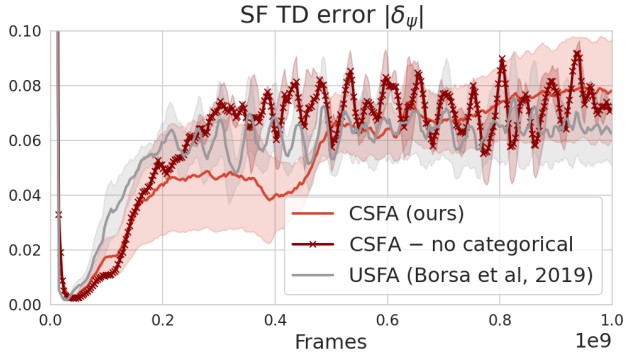

Figure 13: **Evidence that the SF-TD error is less stable for methods that use a point-estimate**. While both scalar and CSFA methods show an increase in error during training, only point-estimate methods (CSFA - no categorical, and USFA) exhibit oscillatory behavior. In contrast, CSFA, which utilizes a probability mass function for return estimation, demonstrates a consistently more stable SF-TD error. We note that all methods increase TD-error as the agent learns because the error is over larger values. In Figure 14, we show that oscillatory behavior correlates with poor GPI performance.

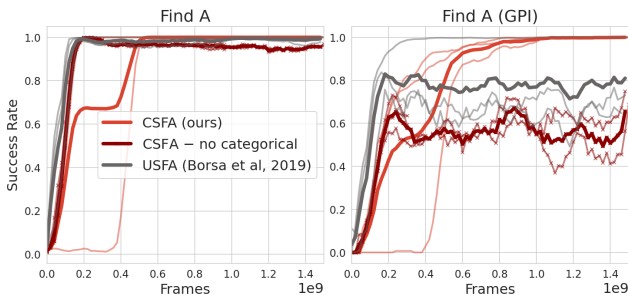

Figure 14: **While all methods can learn Find Tasks, only CSFA can do GPI with find tasks**.

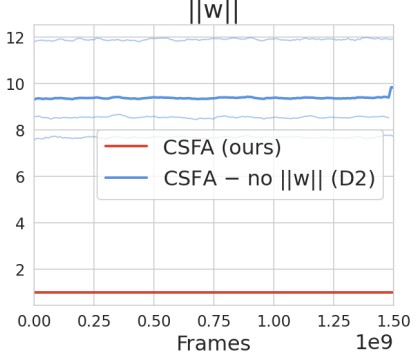

(a) **When we do not bound $w$, it can have a relatively large magnitude**.

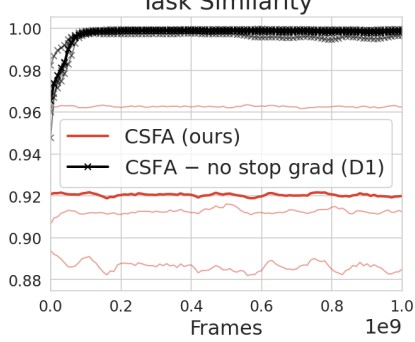

(b) **When we don't stop gradients to $w_\theta$ through Q-learning, task encodings can collapse to a single angle**. We measure the average cosine similarity between pairs of tasks. If we pass gradients from Q-learning (CSFA - no stop grad), tasks collapse onto a single line.

Figure 15: Analysis of task encoding for two different conditions.

## C.2  More ablations

Here we present additional ablation results using the full curriculum in §5.1. We ablate the number of bins used to represent our probability mass function (Figure 16) and we ablate the depth of the residual network used to parameterize cumulants (Figure 17). For both, networks with more capacity (i.e. more bins or more residual blocks) are better able to perform GPI. To be clear, this is not saying that higher capacity networks get better *train performance*, instead this says that higher-capacity networks enable representations that better support SF&GPI.

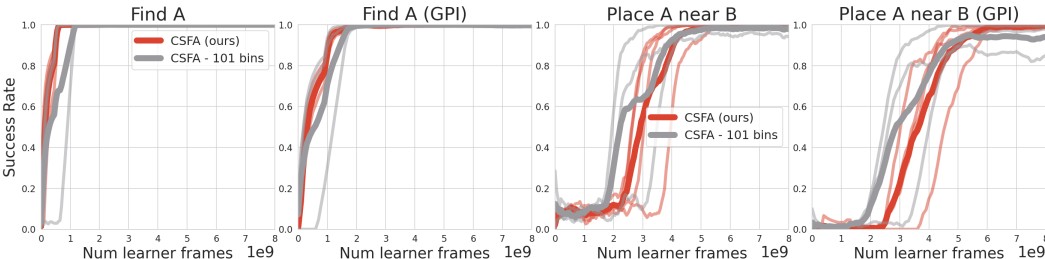

Figure 16: **Ablation on number of bins.** We see that as we decrease the number of bins (i.e. the capacity), the *GPI* performance degrades. We hypothesize that more bins provide more capacity for modelling a wider range of cumulant-returns.

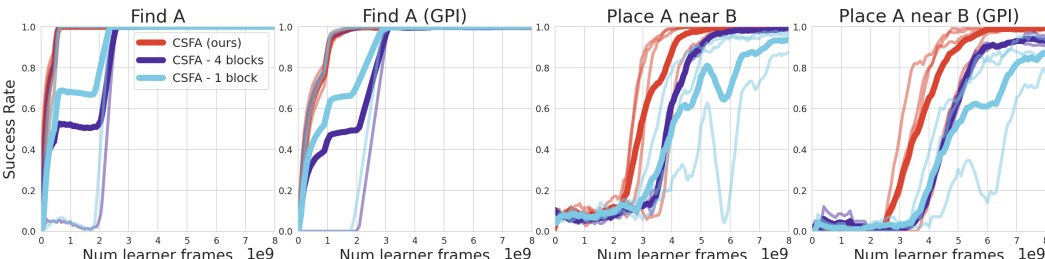

Figure 17: **Ablation on number of ResBlocks.** We see that as we decrease the number of residual blocks, the *GPI* performance degrades for long-horizon tasks. With 4-blocks, performance degrades from 100% to 90%; with 1 block, performance degrades from 95% to 85%. This suggests that use more residuals blocks better supports cumulant discovery for long-horizon sparse-reward tasks.

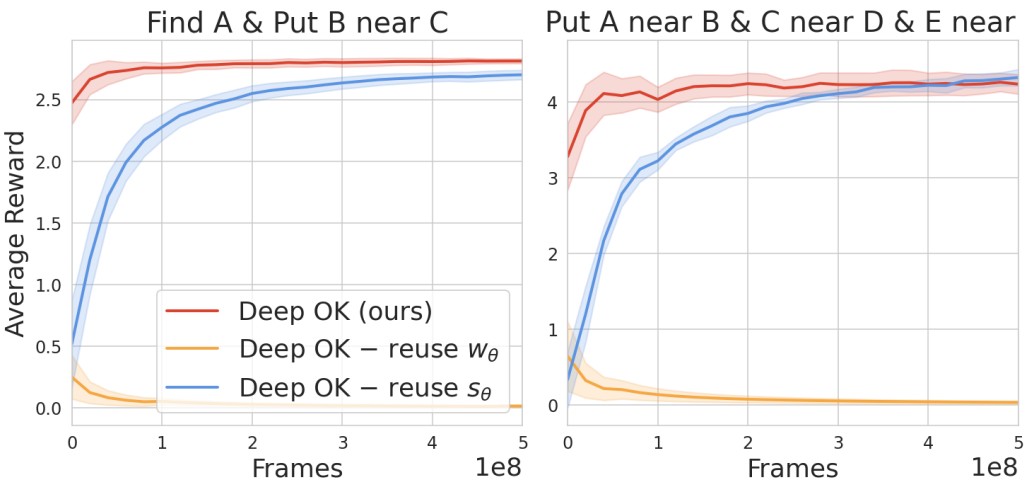

Figure 18: **Ablation on learning a new state function and task encoder.** We see that re-using our state function still enables learning, albeit more slowly than learning a new state encoder for the transfer policy. We see that re-using the task encoder prohibits transfer. We conjecture that this is because this leads to new representations for the train tasks. Since train task encodings parameterize the SF-estimates, this may induce error in the SF-estimates over the course of learning.

# D   Full results

We present both training and GPI performance for the ablation results in §5.1. The original plots only showed GPI performance.

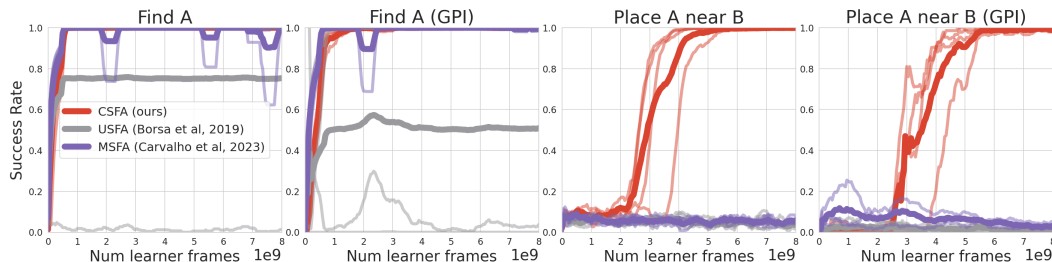

Figure 19: **CSFA discovers representations compatible with GPI across short and long task-horizons**. We see that while most USFA seeds learn Find tasks, a smaller subset can perform GPI. MSFA can do so consistently. Neither are able to learn our longer horizon task. We hypothesize that this is because they approximate SFs with a point-estimate. (4 seeds)

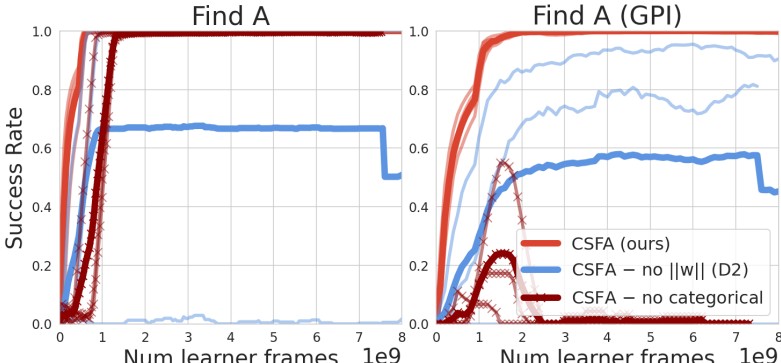

Figure 20: **Full results for short-horizon task ablation**. This plot makes clear that CSFA with a scalar estimator (CSFA - no categorical) gets perfect training performance but terrible GPI performance. It also show that when we don't bound $||w||$, we may sometimes get good train performance but there is still a drop in GPI performance.

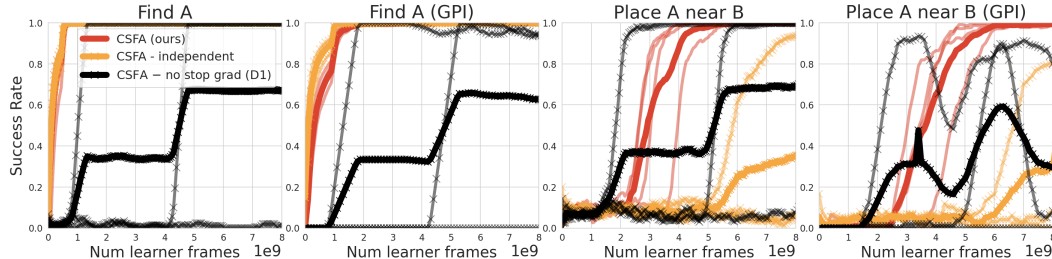

Figure 21: **Full results for long-horizon task ablation**. This plot makes clear that using a separate categorical SF-estimator for each cumulant (CSFA-independent) leads to a drop in GPI performance for long-horizon tasks when successfull (90% to 80%) but is often unsuccessful in learning long-horizon tasks. This is even more striking when we enable gradients to pass from Q-learning into our task encoder (CSA - no stop grad), where multiple seeds will get perfect train performance for long-horizon tasks but exhibit very unstable GPI performance.

# E   Implementation Details

**The code is proprietary so we cannot release it but we included implementation details below to enable reproducibility**.

Everything was implemented with the jax ecosystem [40]. Architectures were implemented with the haiku library. Optimizers were implemented with the optax library. RL algorithms were implemented with the rlax library. All ResNets Blocks correspond use the "BlockV1" definition found in the haiku library. All activation functions used were the relu activation unless otherwise stated.

The rest of this section is structed as follows. In §E.1 we discuss the compute we used in our experiments. In §E.2 we detail SF-based architectures: MSFA, USFA, CSFA. In §E.3 we detail transfer architectures: SFK, Impala (used for MTRL), and Distral. In §E.4 we describe learning for each method. Finally, in §E.5 we describe our hyperparameter search.

## E.1   Compute resources

All experiments were run using Google Dragonfish TPU with $2 \times 2$ topology devices. Experiments for training tasks (§5.1) lasted about 1.5 days. Our simplified experiments in §C.1 lasted 5-6 hours. Transfer experiments (§5.2) lasted 2-3 hours. We ran a learner, actors, and evaluators on TPUs. All experiments were carried out using a distributed A3C setup [30] with discrete actions. Each experiment had 1 learner, 512 actors and 2 evaluators per task, each running in parallel.

## E.2   SF-based architectures: USFA, MSFA, & CSFA

All SF-based architectures had more-or-less the same architecture with the exception of the SF-Approximator. We present a diagram of this general architecture in Figure22. Below we detail this architecture along with differences across methods.

Figure 22: **Architecture schematic used by SF-based methods**.

**Observation function**. All architectures used a ResNet as their observation encoder with channel dimensions $(16, 32, 32)$ for each ResNet block. Each ResNet block had two Resdiual layers and used a stride of 2 throughout. The output of this was then flattened, concatenated with the previous action, and passed through a projection of dimension 256.

**Task encoder**. All architectures embedded words in task descriptions using standard word embeddings of dimension 64. The sequence was then fed through an LSTM of dimension 64. The outputs of each LSTM time-step were then summed and passed through a projection to the cumulant-dimension (16 across all our experiments). THe output was then divided by its L2 norm.

**State function**. All architectures used an LSTM of dimension 512. MSFA differed in that it used 4 LSTMs. We tried having their individual size be 512/4=128 and 1024/4=256 and found much

better performance with a smaller LSTM. They shared information with a transformer attention block. Please see Carvalho et al. [6] for details.

**Cumulant function**. All architectures used a MLP-based ResNet with 8 blocks of dimension 256 each. To create these ResNet blocks, we use the "BlockV1" definition in haiku but replace all convolution layers with linear layers. We then pass the output of the ResNet to a 4-layer MLP where each layer has dimension 256. The MLP outputs scalar cumulants. MSFA differed in that it used a separate ResNet + MLP netwrok for each state module output. The outputs were then concatenated. This again follows Carvalho et al. [6].

**SF Approximators**

- CSFA used an MLP with dimensions [512, 512] and output size A*N. We converted back and forth from the two-hot representation using rlax library functions. In order to re-use this MLP across cumulants, we embed the cumulant dimension with a word embedding of dimension 256. The input to the SF-MLP is then the concatenation of (a) cumulant embedding, (b) the task encoding, and (c) the current state.

- USFA used an MLP with dimensions [512, 512] and output size A*C. The input to the SF-MLP is the concatenation of (a) the task encoding, and (b) the current state.

- MSFA used 4 MLPs with dimensions [512, 512] and output size A*C/4. The outputs were then concatenated to the form the full set of SFs. The input to each SF-MLP is the concatenation of (a) the task encoding, and (b) the current module-state. We refer to [6] for a detailed implementation notes.

A=number of actions, C=number of cumulants, N=number of bins. CSFA used $N = 301$ bins evenly spaced between $[-5, 5]$. All methods used $C = 16$ cumulants and Playroom has $A = 46$ actions.

## E.3  Transfer architectures

All transfer methods were based off of an Impala architecture because it has been used in previous Playroom resuts [31, 32, 10]. Impala and Distral have more-or-less the same architecture, except that Distral has an additional centroid policy head. Below we share functions shared across methods. In §E.3.1, we detail implementation specific to Impala and Distral and in §E.3.2, we detail implementation specific to SFK. We present diagrams of these architectures in Figure23.

**Task encoder**. We embed words in task descriptions using word embeddings of dimension 64. The sequence was then fed through an LSTM of dimension 64. The outputs of each LSTM time-step were then summed. This sum is the output.

**Value function**. All methods use an MLP with 512 hidden dimension and 1 scalar output.

### E.3.1  Impala & Distral

**Observation function**. All architectures used a ResNet as their observation encoder with channel dimensions $(16, 32, 32)$ for each ResNet block. Each ResNet block had two Resdiual layers and used a stride of 2 throughout. The output of this was then flattened, concatenated with (1) the previous action and (2) the task. This is then passed through a projection of dimension 256.

**State function**. All architectures used an LSTM of dimension 512.

**Policy**

- Impala uses an MLP with dimensions [1024] and output size A. The policy is parameterized with a softmax.

- Distral learns two functions: one centroid function produces logits $h(a|s)$ and a task-specific function produces logits $f_i(a|s)$. The task-policy is then parameterized as $\pi(a|s) \propto (\alpha * h(a|s) + \beta f_i(a|s))$. Following Teh et al. [22], we set $\alpha = .5$ and $\beta = 1$ for acting. During learning we use a different $\beta$ to encourage distillation. We detail this in §E.4.

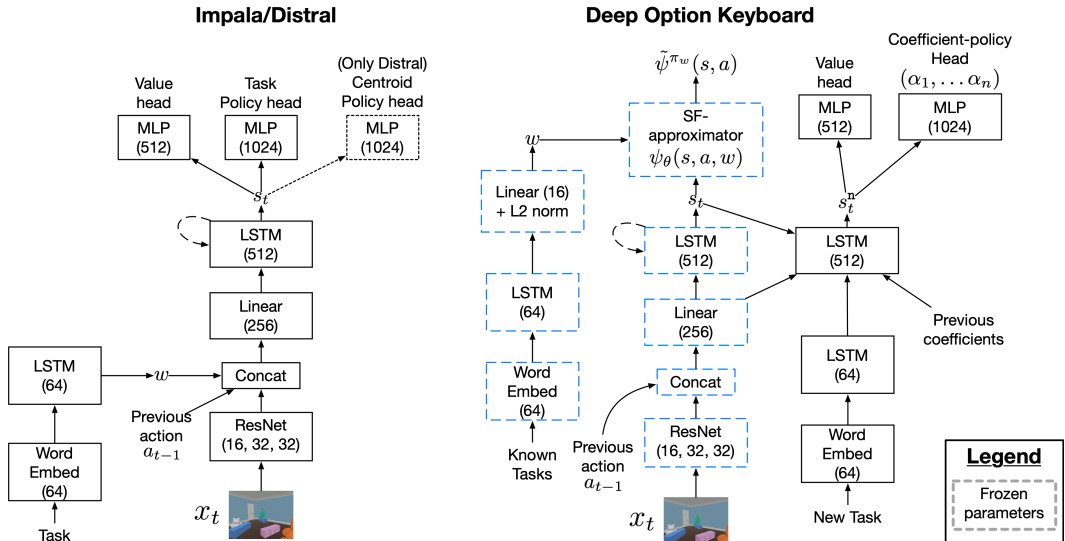

Figure 23: **Architecture schematic used by transfer methods**.

### E.3.2 SFK

SFK uses the CSFA architecture but freezes all parameters. It then learns a new state function, task encoder, policy head, and value function. In principle, more could be re-used and potentially enable even faster learning but we opted for simplicity.

**New state function**. We use LSTM of dimension 512 that takes in the state from CSFA's LSTM, the output of the observation encoder, and previous coefficients produced by the coefficient-policy.

**Policy**. SFK uses an MLP with dimensions [1024] and output size W, where W="number of training tasks" to use for GPI. In our experiments, W was 32.

### E.4 Learning

All trainable parameters were updated with the **adam optimizer** [41]. For each method, we set the following adam hyperparameters: $\beta_1 = 0$, $\beta_2 = .95$, $\epsilon = 6e{-}6$.

### E.4.1 CSFA, MSFA, & USFA

Following Mnih et al. [26], all architectures had a replay buffer that held up to 100,000 trajectories of length 30. This is an *off-policy* setting. Following Mnih et al. [26], all methods used target network parameters $\theta^o$ that were updated smoothly using the online network parameters $\theta$ with a coefficents of .9. That is, at iteration $i$, $\theta_i^o = \theta_{i-1}^o * .1 + .9 * \theta_i$. All methods used a learning rate of $3e{-}4$ and set the max gradient norm to .5. We used the following coefficients to balance loss terms: $\beta_Q = 1e3$, $\beta_\phi = 1e4$. Since CSFA uses a categorical loss $\mathcal{L}_\psi^{\mathtt{cat}}$ to learn SFs, which has a different magnitude from the scalar loss used to learn Q-values and , we found that we needed to make this smaller with $\beta_\psi = 8e{-}3$ whereas we used $\beta_\psi = 5e2$ for scalar-based methods.

### E.4.2 SFK, Impala, & Impala

We collected training data using a FIFO replay queue with 36 trajectories of length 128. Batches of 32 trajectories were sampled uniformly. This was an *on-policy* setting. SFK used a learning rate of $8e{-}5$. SFK learns to select actions with the Impala learning algorithm, but with an augmented actions-space which produced task-coefficents instead of environment actions. Impala used a learning rate of $3e{-}4$ for training and $8e{-}5$ for transfer. Distral used a learning rate of $8e{-}6$ for training and $3e{-}5$ for transfer. All methods set the max gradient norm to $40$.

For Distral, we set $1/\beta = 3e-3$. We set $\alpha = .5$. Since Distral is a policy-gradient method, we added its policy distillation loss $\mathcal{L}_{\texttt{dist}}$ to an Impala loss $\mathcal{L}_{\texttt{impala}}$. The full loss was $\mathcal{L}_{\texttt{distral}} = \mathcal{L}_{\texttt{impala}} + \alpha_{\texttt{dis}}\mathcal{L}_{\texttt{dist}}$. We used $\alpha_{\texttt{dis}} = 5.0$ for training tasks and $\alpha_{\texttt{dis}} = .1$ for transfer tasks.

## E.5 Hyperparameters

All methods and their corresponding hyper-parameters are based on prior work that has learned in Playroom environment [32, 42, 31]. Specifically, we mainly used hyperparameters from [31] since they also study how Impala-based MTRL can transfer in Playroom (though they do so in a continual learning setting with skewed distributions). Starting from their hyperparameters, we swept the following.

Impala (MTRL). Prior hyperparameters worked well here since we used them for training tasks. For transfer, we swept learning rates $[3e-4, 8e-5, 3e-5, 8e-6, 3e-6]$, value hidden sizes [200, 512, 1024], and policy hidden sizes [200, 512, 1024].

Distral was originally built on top of the asynchronous advantage actor critic (A3C) algorithm [43]. We built on top of Impala since it is a more advanced actor-critic algorithm that has been studied in playroom. As such, we kept hyper-parameters and tuned the following: (1) learning rates $[3e-4, 8e-5, 3e-5, 8e-6, 3e-6]$, (2) $\alpha_{\texttt{dis}}$, $[10, 5, 1, .1, , 01, .001]$, (3) Impala entropy coefficent $[9.4e-3, 9.4e-4, 9.4e-5, 9.4e-6, 9.4e-7]$, (4) $1/\beta$ $[3e-3, 1e-3, 3e-4]$.

Value-based methods (as SF-based methods) are more challenging to learn for Playroom. To facilitate search, we concentrated our search on simpler "Find" tasks. First, we searched for MSFA which has shown good performance with discovered cumulants. Reusing optimizer settings from Impala, we set $\beta_\psi = \beta_\phi = 0$ and searched over (1) $\beta_Q$: [1, 1e1, 1e2, 1e3, 1e4] (2) max grad norm: $[40, 5, .5]$. Once agents could learn Find tasks, we searched over (3) resblocks: [1, 4, 8] (4) $\beta_\phi$: [1, 1e1, 1e2, 1e3, 1e4] and (5) $\beta_\psi$: [1, 1e1, 5e1 1e2, 5e2, 1e3]. We found the same hyper-parameters worked well for USFA and CSFA. For CSFA, we searched $\beta_\psi$: [.8, .3, .08, .03, .008, .003, .0008] and bins [101, 301]. We began with $\beta_\psi$ values which led the SF-loss to be approximately equal to the Q-value loss and then went downward from there since the SF-loss can be thought as a regularizer on the Q-value loss.

For SFK, we were able to re-use Impala HPs but searched over max grad norm $[40, 5, .5]$, value hidden sizes [200, 512, 1024], policy hidden sizes [200, 512, 1024], and Impala entropy coefficent $[9.4e-3, 9.4e-4, 9.4e-5, 9.4e-6, 9.4e-7]$.

# F    Environment Details

This work is a significant increase in complexity from prior successor feature literature [3, 4, 8, 6, 1]. The most complex task studied by prior work were "go to" tasks in the DMLab environment by [8]. Here, no object-interaction is required. (1) Playroom "place near" tasks add object-interaction, while increasing the task horizon and reward sparsity (2) prior work only combined 4 "go to" tasks while we combine 8 Find and 24 place nears tasks (3) prior work used DMLab with 8 actions [44] while Playroom has 46 actions. We have added this discussion to the main text.

## F.1    Observation space

The agent experiences (1) observations as pixel-based RGB images that have dimensions $72 \times 96 \times 3$ and (2) task descriptions as string descriptions in a synthetic language.

## F.2    Action Space

The agent can rotate its body and look up or down. To pick up an object it must move its point of selection on the screen. When it picks up an object, it must continuously *hold it* in order to move it elsewhere. To accomplish this, the agent has 46 actions. Objects are helf as long as the agent is emitting a "GRAB" action, and dropped in the first instance that a GRAB action is not emitted. We describe these actions in Table2. An action-repeat of 4 was applied (i.e. each action was repeated 4 times).

Table 2: 46 actions available to the agent. The number next to each action name indicates how many meters the agent will move with that action. For example, move forward(1) indicates that agent will move forward 1 meter.

| Type of action | actions | | |
|---|---|---|---|
| | | | |
| **movement without grip** | noop | move forward(1) | move forward(-1) |
| | move right(1) | move right(-1) | look right(1) |
| | look right(-1) | look down(1) | look down(-1) |
| **fine-grained movements** | move right(0.05) | move right(-0.05) | look down(0.03) |
| **without grip** | look down(-0.03) | look right(0.2) | look right(-0.2) |
| | look right(0.05) | look right(-0.05) | |
| **movement with grip** | grab | grab + move forward(1) | grab + move forward(-1) |
| | grab + move right(1) | grab + move right(-1) | grab + look right(1) |
| | grab + look right(-1) | grab + look down(1) | grab + look down(-1) |
| **fine-grained movements** | grab + move right(0.05) | grab + move right(-0.05) | grab + look down(0.03) |
| **with grip** | grab + look down(-0.03) | grab + look right(0.2) | grab + look right(-0.2) |
| | grab + look right(0.05) | grab + look right(-0.05) | |
| **object manipulation** | grab + spin right(1) | grab + spin right(-1) | grab + spin up(1) |
| | grab + spin up(-1) | grab + spin forward(1) | grab + spin forward(-1) |
| | grab + pull(1) | grab + pull(-1) | |
| **fine-grained object** | grab + pull(0.5) | grab + pull(-0.5) | pull(0.5) |
| **manipulation** | pull(-0.5) | | |

## F.3    Training Tasks

The agent experiences $n_\kappa = 72$ training tasks $\mathbb{T}_{\texttt{train}} = \{\kappa_1, \ldots, \kappa_{n_\kappa}\}$ composed of "Find A" and "Place A near B". $|A| = 18$ and $|B| = 3$.

The set $A$ comprises 18 items that can be picked up: boat, bus, car, helicopter, keyboard, plane, robot, rocket, train, racket, candle, mug, hairdryer, picture frame, plate, potted plant, roof block, and rubber duck.

The set $B$ comprises 3 items that can be stacked: book, cube block and sponge.

"Find" tasks follow the form "Find a A", where the placeholder "A" represents the object to be found. "Put near" tasks are of the form "Put a A near a B", where "A" represent the object to be picked

up and "B" represents the object that A should be placed near. All methods experienced the same training curriculum. For all tasks, the agent only get reward upon completing all subtasks. All tasks provide a reward of 1 upon-completion.

## F.4 Transfer Tasks

Transfer tasks were made using a subset of $8$ pickupable objects $A'$: boat, bus, car, helicopter, keyboard, plane, robot, rocket.

Transfer tasks where then combinations of known train tasks that used $A'$. An "and" strong was placed into the task description to indicate combinations. For example "Place A near B and C near D and E near F" would be "place a boat near a book **and** place a bus near a cube block **and** place a keyboard near a sponge". Objects were sampled with replacement, indicating that an object may appear multiple times but would have a different color. For all tasks, the agent only get reward upon completing all subtasks. "Find" subtasks provide a reward of 1 and "Place" subtasks provide a reward of 2. Returning to our example, since it has 3 subtasks, it would provide a reward of 6 but only upon completion of *all* 3 subtasks.

