# OpenReview forum: "Combining Behaviors with the Successor Features Keyboard"
_NeurIPS.cc/2023/Conference — NeurIPS 2023 poster_

### Official Review · Reviewer_cEsx · 2023-06-27

**Soundness:** 2 fair
**Presentation:** 2 fair
**Contribution:** 2 fair
**Rating:** 5
**Confidence:** 3

**Summary:**

The author(s) present a novel and efficient approach to knowledge transfer in long-horizon tasks, aiming to minimize interactions with the environment (less frames required in training). Their proposed method leverages Categorical Successor Feature Approximator (CSFA) within an enhanced deep OK network, resulting in several tasks (e.g., Find, Place).

**Strengths:**

- Proposed  a dynamic transfer query that enables sharing among task-encoding and SF-approximators;
- Implemented this methodology in long-horizon tasks.


**Weaknesses:**

1. The authors did not clearly position their contribution in relation to other approaches commonly used in long-horizon tasks, such as hierarchical approaches, hindsight approaches and meta-learning approach [1-3].
2. While the overall results of the experiments are significant (e.g., Figure 3 improve 0.2 ~ 0.8 in success rate), the description and documentation of some experiments might be unclear or insufficient. 1) USFA or MSFA might require fewer computational resources compared to MSFA, as their calculations appear to be simpler (line 246). However, without further detailed information, it is difficult to make a definitive judgment. I suggest that the authors include memory and computational load comparisons among these methods. 2) The author's comparisons are focused on MTRL, but there is a lack of comparison with other approaches mentioned in references [1-3]. It would be beneficial to include comparisons with other relevant approaches mentioned in reference [3].
3. From the paper, it is unclear why MSFA is considered sharable or suitable for fitting different tasks. How it compare with other pre-training encodings or other method representations.
4. Some typos, e.g., "Transfer tasks are conjunctions of known known tasks", line (289)

[1] Long-Horizon Visual Planning with Goal-Conditioned Hierarchical Predictors, NeurIPS 2020

[2] Learning to Reach Goals via Iterated Supervised Learning, NeurIPS 2020

[3] Skill-based Meta-Reinforcement Learning, ICLR 2022

**Questions:**

see weakness and limitations

**Limitations:**

- This paper could benefit from an expanded evaluation of the transfer learning quality beyond just the success rate metric, e.g. adding visualizations of the baseline task encoding and dynamic query encoding to show the benefit of dynamic query is scalable.
- This study lacks of discussion about the upper bound of the number of tasks it can effectively handle during sharing coding. The author primarily tests the approach on two main tasks ("find" and "place") with several sub-tasks or steps required to complete each main task. However, it remains unclear what the upper limit number of main task and sub-tasks?

---

> ### Author Rebuttal · Authors · 2023-08-09
>
> Thank you for this review. Below we respond to specific questions.
>
> **The authors did not clearly position their contribution in relation to other approaches commonly used in long-horizon tasks, such as hierarchical approaches, hindsight approaches and meta-learning approach [1-3]. The author's comparisons are focused on MTRL, but there is a lack of comparison with other approaches mentioned in references [1-3]. It would be beneficial to include comparisons with other relevant approaches mentioned in reference [3]**
>
> Thank you for pointing out this work. We have included a discussion on methods for learning long-horizon tasks in the paper and include these 3 papers. In summary, [1,3] assume access to offline datasets with useful (though potentially sub-optimal) behavior, and [2,3] demonstrate results on environments with hand-designed state-representations. In our work we don't assume offline datasets and aim to learn state and goal representations from partial observations of a pixel-based 3D environment. We do not have the resources to implement and compare to these methods in this rebuttal phase.
>
> [1] learns long horizon tasks via hiearchical planning, whereas we focus on transferring value functions via successor features. [1] assumes access to diverse (but potentially sub-optimal) demonstration data that is used to learn a dynamics model used for goal-directed planning. Our method does not assume any demonstration data. Thus, [1] is not applicable to our setting without augmenting it with a method to actively explore the environment to produce data for learning its dynamics model.
>
> [2] learns a goal-conditioned policy for data that is relabeled such that the final state in a trajectory is a re-defined as a goal state for a self-imitation learning objective. While this method is applicable to our setting, there are several differences between settings we study. First, they assume access to a hand-designed state-representation which also defines the goal-space. This is in contrast to our setting where an agent must learn state and goal representations from high-dimensional, pixel-based, partial observation of a 3D environment. This challenge is exacerbated by the environment's large state space that grows exponentially with the number of objects. Our tasks each have task objects and at least 2 distractor objects. In summary, it is not clear that [2] is applicable to our setting without modifications to accodomate (1) learning state- and goal-representations in a (2) partially observable, pixel-based 3D environment with (3) a large state-space induced by task and distractor objects.
>
> [3] learns long-horizon tasks by integrating meta-reinforcement learning with a hierarchical policy that leverages skills that are obtained from an offline dataset. In our work, we do not assume access to a dataset that has useful skills to be learned. Additionally, [3] studies environments with hand-designed, symbolic state representations whereas we focus on learning state representations for a partially observable, pixel-based, 3D environment.
>
> **USFA or MSFA might require fewer computational resources compared to CSFA, as their calculations appear to be simpler (line 246). However, without further detailed information, it is difficult to make a definitive judgment. I suggest that the authors include memory and computational load comparisons among these methods.**
>
> Our interpretation of this comment is that you are referring to the loss function of CSFA (eqs 6 & 7) vs. the loss function of USFA/MSFA (eq 4), (line 146 not 246). Please see the joint response for this.
>
> **From the paper, it is unclear why MSFA is considered sharable or suitable for fitting different tasks. How it compare with other pre-training encodings or other method representations.**
>
> We consider MSFA because it (and USFA) are the only methods designed to share an SF-estimator across different task encodings, i.e. to learn $\psi_{\theta}(s,a,w)$. Other methods learn separate estimators $\psi^{\pi_i}_{\theta_i}(s,a)$ with different parameters $\theta_i$ for each task encoding $w_i$. We have clarified this in the text.
>
>
> **This study lacks of discussion about the upper bound of the number of tasks it can effectively handle during sharing coding. The author primarily tests the approach on two main tasks ("fìnd" and "place") with several sub-tasks or steps required to complete each main task. However, it remains unclear what the upper limit number of main task and subtasks?**
>
> While we do now know the upper limit on the number of main tasks our method can handle, we emphasize that this work is a significant increase in complexity from prior work in the literature[4-8]. We discuss this in more detail in our shared response above. We note that prior work focused on combining 4 "go to" tasks. If one combines 2 "go to" tasks, this leads to $\left(\begin{array}{l}4 \\\\ 2\end{array}\right)=6$ combinations. We have 8 find tasks and 24 place near tasks. If we combine just 2 place near tasks, this leads $\left(\begin{array}{l}24 \\\\ 2\end{array}\right)=276$ possible transfer combinations. While we did not explicitly study increasing the number of find and place tasks, we are optimistic that one can scale the capacity of a sufficiently expressive task encoder with the number and complexity of tasks being learned.
>
>
>
> [1] Long-Horizon Visual Planning with Goal-Conditioned Hierarchical Predictors
>
> [2] Learning to Reach Goals via Iterated Supervised Learning
>
> [3] Skill-based Meta-Reinforcement Learning
>
> [4] Successor Features for Transfer in Reinforcement Learning
>
> [5] Transfer in Deep Reinforcement Learning Using Successor Features and Generalised Policy Improvement
>
> [6] Universal Successor Features Approximators
>
> [7] Composing Task Knowledge with Modular Successor Feature Approximators
>
> [8] The Option Keyboard Combining Skills in Reinforcement Learning

---

> > ### Comment · Reviewer_cEsx · 2023-08-15
> >
> > I thank the authors for clarifying my questions. I am satisfied with the author's rebuttal. Since the author has partially addressed my concerns, I would like to raise one point. I recommend that the authors add computation complexity and task limitations to the paper.

---

### Official Review · Reviewer_xth2 · 2023-07-05

**Soundness:** 3 good
**Presentation:** 3 good
**Contribution:** 2 fair
**Rating:** 5
**Confidence:** 3

**Summary:**

This paper focuses on the Option Keyboard (OK) task which was recently proposed as a method for transferring behavioral knowledge across tasks. They propose a new method of Deep Option Keyboard (Deep OK), which enables transfer with discovered state features and task encodings. To enable discovery, they further propose the Categorical Successor Feature Approximator (CSFA), a novel learning algorithm for estimating SFs while jointly discovering state features and task encodings. With Deep OK and CSFA, this paper achieves the first demonstration of transfer with SFs in a challenging 3D environment where all the necessary representations are discovered. The experiment results show that CSFA can discover SFs compatible with SFGPI and Deep OK achieves strong jumpstart performance for transfer to combinations of training tasks.


**Strengths:**

1.	Transfering with SFGPI in a large-scale multi-task setting, discovering cumulants and task encodings, while sharing a task encoder and SF approximator across tasks gives some insights to the community and seems novel.
2.	The paper’s writing is generally clear and the figures were helpful for understanding.
3.	The problem formulation is clear and the approach is well-reasoned.
4.	The experiments are sufficient and convincing to some extent.


**Weaknesses:**

1.  Though SFGPI is dynamic and can share a task encoder and SF approximator across tasks, its generalization ability is not clear and no more discussion can be found in the paper. Maybe a more general method that sampling queries from the task encoding space that is more fairly concentrated could be designed and at least provide some discussions.
2. In Figure 3, CSFA discovers representations compatible with GPI across short and long task horizons. The longer horizon is defined by just one “Place Near” task. Will CSFA outperform MSFA and USFA on other general tasks?
3. Deep Ok doesn’t reach an optimal performance but uses far fewer frames compared with Distral.


**Questions:**

In Line 364, the author claims that  “This may enable other methods that leverage SFs in a multi-task setting”. Since Deep OK can’t get the optimal performance, how can leverage SFs for discovering representations and get the optimal performance simultaneously？

**Limitations:**

See the weakness and question sections.

---

> ### Author Rebuttal · Authors · 2023-08-09
>
> Thank you for your review. We are glad that you found the writing clear and approach well-reasoned. Below we respond to specific questions.
>
> **Maybe a more general method than sampling queries from the task encoding space that is more fairly concentrated could be designed and at least provide some discussions.**
>
> While our analysis indicated that the learned task encoding space was somewhat concentrated, at the scale and complexity of our problem, we did not find this to be a bottleneck. In Figure 7, we compare againt sampling from a gaussian distribution (Deep OK - binary) and we show that this does not generally learn as quickly.
>
> If future work does encounter this challenge, one solution might be to leverage a contrastive learning framework similar to CLIP [1] to learn task encodings that are better dispersed (for example, on a hyper-sphere). CLIP has been successfully applied to learning observation representations that are aligned with text representations. An agent's state representation is a representation of their stream of visual observations, and when goals are given via text, they are text representations. Thus, given the success of CLIP, it is reasonable that future work can leverage such a contrastive learning method to improve the quality of the task encoding space. We have added this to our discussion section.
>
> **The longer horizon is defined by just one “Place Near” task.** **Will CSFA outperform MSFA and USFA on other general tasks?**
>
> We believe that the benefits of our approach will be general; indeed, we focus on "place near" tasks because they are substantially more complex and difficult than the tasks considered in prior work, which focused on simpler "go to" tasks [3-7]. We discuss this in more detail in the shared response above. Given the performance improvement of CSFA on these more complex tasks, nothing suggests it shouldn't continue to improve in other settings with similar difficulty.
>
> **Since Deep OK can’t get the optimal performance, how can leverage SFs for discovering representations and get the optimal performance simultaneously?**
>
> One could improve/build on Deep OK to get optimal performance by integrating it with a distillation method. The main benefit of Deep OK (and SF&GPI more generally) is that it enables relatively good performance, relatively quickly. One can leverage the policy learned by Deep OK $\pi^{\tt SF}$ as a source policy in a policy distillation algorithm that transfers its knowledge to a fully parametric action policy $\pi^{\tt target}$. For example, by minimizing the KL-divergence between the two policies as the target policy learns to maximize reward, as with Kickstarting RL [2]. One could design this algorithm so that it trades off between selecting actions from $\pi^{\tt SF}$ to selecting actions from $\pi^{\tt target}$ over the course of learning. We have added this to our discussion section.
>
> We emphasize that SF&GPI is not intended to provide optimal transfer performance. Instead, they are intended to provide reasonably good, fast transfer performance. See [3-5] for more.
>
> [1] Learning Transferable Visual Models From Natural Language Supervision
>
> [2] Kickstarting deep reinforcement learning
>
> [3] Successor Features for Transfer in Reinforcement Learning
>
> [4] Transfer in Deep Reinforcement Learning Using Successor Features and Generalised Policy Improvement
>
> [5] Universal Successor Features Approximators
>
> [6] Composing Task Knowledge with Modular Successor Feature Approximators
>
> [7] The Option Keyboard Combining Skills in Reinforcement Learning

---

### Official Review · Reviewer_RZN8 · 2023-07-05

**Soundness:** 4 excellent
**Presentation:** 3 good
**Contribution:** 4 excellent
**Rating:** 8
**Confidence:** 4

**Summary:**

This work focuses on attempting something which no previous work has done before;  learning task encodings, cumulants and successor features (SFs). The author(s) proposed a solution based on learning these three functions jointly. They considered a model that learns the SFs as probability mass function using the two-hot representations inspired from MuZero. They named this model as the Categorical Successor Feature Approximator (CSFA). With the learned SFs, the authors will then be able to perform Generalised Policy Iteration (GPI) to obtain the policy for the relevant task at evaluation phase. In order to find a better policy for this task, the authors also introduced a method named Deep Option Keyboard (DOK), which relies on a parameterised function to output coefficients which are then used as preference vectors. The models are evaluated in a 3D environment, using pixel observations. The tasks included going to certain locations and collecting and then dropping objects at different locations. In particular, DOK is used to study the effects of transfer in sparse reward and long horizon tasks. The proposed model is able to perform better than the baselines such as Universal Successor Features and Modular Successor Feature Approximators.


**Strengths:**

I find the work interesting as this is an attempt at a challenging problem of learning task encodings, cumulants and successor features jointly. The figures, in particular, figure 2, is clear in aiding the reader to understand the architecture of their model. The writing overall was mostly clear and easy to read. The experiment's descriptions are concise and helped me to understand the setup a lot even though I am not familiar with the playroom environment. The authors also performed ablation studies which showed that the key to their model is learning the SFs using a probability mass function, rather than the canonical expected values.



**Weaknesses:**

Although there are no major weaknesses in my opinion, I do have some points for thought.

Firstly,I am not sure it is a good idea to call the mechanism for learning the preferences, “Deep Option Keyboard” when it seems that there are no options involved.

Secondly, it would also perhaps be a good idea to see how this model compares with the standard Q-learning algorithm, such as Rainbow or even just a distributional RL model since the proposed model uses a probability mass function to get an idea how much of the transfer benefits one could achieve from the utilization of the successor features.

Thirdly, there was some mention of static vs dynamic query. Perhaps it will be helpful to give a clear definition of what it means by static and dynamic query.


**Questions:**

Q1. The authors discretize the cumulant returns values using bins. Is there a study done to show how the different number of bins impact the performance of the agent?

Q2. I understand that Bernoulli distribution was used as the parameterized probability function. What are the other distributions that were considered? Why did Bernoulli distribution do better than the others? Do you have any insight on this?

Q3. On line 273, the authors mentioned that sharing an estimator across cumulants is necessary. What do you mean by keeping categorical representation? Is this successor features or state function or something related to cumulant network (with reference to figure 2).


(Potential typos):
Line 289: Known known tasks?

Caption of figure 5: Distral and MTRL learn an ??? about the same speed. My guess is “an optimal or policy”?

---

> ### Author Rebuttal · Authors · 2023-08-09
>
> Thank you for this positive review. We are happy you appreciated this submission. Below we respond in detail to your questions.
>
> **Changing name to not give false impression of using options.**
>
> Thank you for pointing this out. We have changed the name to "Successor Feature keyboard".
>
> **Adding a Q-learning baseline**
>
> Thank you for this suggestion. USFA, MSFA have both been shown to outperform Q-learning baselines for transfer and IMPALA-based MTRL has been shown to outperform Q-learning baselines in Playroom [1]; given that our approach outperforms all of them, we believe it would outperform Q-learning even more strongly. Unfortunately, we do not have the resources to implement and tune this baseline during this rebuttal period. We have included a discussion of this in the main text.
>
> **Making explicit the difference between static and dynamic query**
>
> Great suggestion. We have made this more explicit in text. We summarize this here: a "static query" for transfer task $\kappa'$ corresponds to the output of $w_{\theta}(\kappa')$ whereas a "dynamic query" is the output of the state-dependent function $g_{\theta}(s, \kappa')$.
>
> **Is there a study done to show how the different number of bins impact the performance of the agent?**
>
> We ablate the number of bins in Figure 14 of the appendix. We find that using more bins is helpful, though fewer bins does reasonably well.
>
> **What kinds of distributions were considered? Why did a Bernoulli distribution do better?**
>
> In addition to a Bernoulli distribution, we also considered a gaussian distribution as the original Option Keyboard did. We showed results for this in Figure 7 (Deep OK - binary). We did not consider other distributions since a Bernoulli distribution is simple to implement and it provided good results. We also studied auto-regressive variants of both distributions and found no improvements.
>
> We believe that a Bernoulli distribution performed well because it facilitates exploiting SF&GPI. SF&GPI enable transfer to linear combinations of task encodings. By leveraging a Bernoulli distribution for sampling task encoding coefficients, the dynamic query is a linear combination of learned task encodings. This is precisely the type of transfer task encoding that SF&GPI has been shown to work well with. We have added a discussion on this to the "discussion and conclusion" section.
>
> **On line 273, the authors mentioned that sharing an estimator across cumulants is necessary. What do you mean by keeping categorical representation? Is this successor features or state function or something related to cumulant network**
>
> Thanks for this clarifying question. We consider estimating $n$ SF with pmfs. Each pmf has a domain of $M$ values $\mathbb{B}=\\{b_1, \ldots, b_M\\}$. We have two choices for the $k$-th SF:
>
> 1. CSFA: Use the same parameters but have an embedding of the dimension $e_k$ be input, i.e. $p_{\psi_k} \propto \exp \left(l_\theta^\psi\left(s_t, w, e_k\right)\right) \in \mathbb{R}^M$
> 2. Ablation: Use separate parameters $\theta_k$, i.e. $p_{\psi_k} \propto \exp \left(l_{\theta_k}^\psi\left(s_t, w\right)\right) \in \mathbb{R}^M$
>
> The setting of this ablation is what is typically done when modeling SFs. When we say "keeping the categorical representation", we simply mean that we still model each SF with a pmf as apposed to using a scalar representation. We have clarified this in the text.
>
> **Caption of figure 5: Distral and MTRL learn an ??? about the same speed.**
>
> We meant "Distral and MTRAL learn **at** about the same speed"

---

> > ### Comment · Reviewer_RZN8 · 2023-08-15
> >
> > Thanks for the rebuttal. I am satisfied with the answers given and have read the reviews and rebuttals from and given to other reviewers. I have no further questions.

---

### Official Review · Reviewer_z7ra · 2023-07-10

**Soundness:** 3 good
**Presentation:** 2 fair
**Contribution:** 3 good
**Rating:** 4
**Confidence:** 3

**Summary:**

First, I must emphasize that I was new to the RL community before this NeurIPS. This is the first RL paper I have reviewed. I have been struggling to understand this paper. I tried my best. I hope the AC, reviewers, and authors may think me helpful.

This paper studies how to transfer a pre-learned RL system to new tasks using successor features and the optional keyboard. Compared with previous methods, new things are as follows: 1) the Q-value is no longer linear about the task encoding, here, the authors use a network $g_{\theta}$ to predict the weight of SF features based of task encoding and state representation. 2) The SR approximation is modeled as the expectation of a discrete distribution, the authors claim this can help stabilize the training. 3) The learning objective is modified. The new objective consists of three parts a) a term to decrease the prediction error of reward using $\phi$, b) a term to decrease the prediction error of reward using the difference of $\psi$, and c) a term to maximize the possibility of current $\psi(a_t,s_t,w)$ to occur using the twohot operation. 4) A new transfer scheme that learns a new task encoder and state function at transfer time, where the task encoding is the weighted sum of a Bernoulli random variable and is aware of the time. The proposed method achieves better performance than USFA and MSFA in 3D playroom dataset, showing a better success rate.



**Strengths:**

# Novelty
Above all, I may not be able to judge the novelty of an RL paper. So a comment based on my experience in my familiar domain.

1. The new assumption of SF and Q-Value: It is good to introduce $g_{\theta}$, but the Q-value is still a linear combination of the SF. So I think this is still somewhat narrow.
2. New SF approximation: In my opinion, the key advantage is that the distribution distributes the difference between $\psi(a_t,s_t,w)$ and $\psi(a_{t+1},s_{t+1},w)$ to $m$ different points. So each point only needs to change modestly when the time progress, and the similar features among different times can be better stored by those points of high density among times. Correct me if I am wrong here. This seems a good point, but the authors lack enough explanation here.
3. New Learning Objective: I don't understand why the authors omit $\mathcal{L}_{\psi}$ in training, the new objective seems to have no term to align the difference of $\psi$ and the cumulant $\phi$. The authors also do not provide enough explanation to why this objective is designed, nor any ablations to explain this.I have a hard time understanding this part.
4. New transfer scheme: This seems a novel point. The authors again use distributions to model the task encoding. This form may be of stronger approximation ability.


# Clarity
1. I like the Summary of Challenges in Fig.8, it helps quickly figure out the clues of this work.
2. The authors give enough reference to key concepts, this helps me a lot in reviewing this paper.



# Significance
Good in the showed case, but narrow in the scenario.

**Weaknesses:**









# Clarity:
One of my major concerns is the clarity of this work. It seems like it was done in a hurry. The notations are used carelessly, usually not unified. I can see the same symbols with different superscripts or subscripts, or arguments. The components of the proposed method lack enough explanation of why they are helpful. Those make me restless, can't sleep overnights and recall them in dreams. I suspect several places use wrong notations, correct me if I am wrong. Some issues of clarity are as follows:
1. Line 112 you miss an "of".
2. It is $r_t^{\tau}$ or $r_t$ in the second eq. of Eq (4)?
3. I recommend to write Eq. (7) as $\mathrm{Twohot}(y_t^{\psi_k})\log p_{\psi_k}^T$ for clarity.
4. Not every fig of this paper has an error bar.
5. Notations are a bit confusing and frequently unexplained. I have a hard time understanding them.
6. What does $\theta^o$ mean in Eq. (6\&7)?
7. Righthand side of Eq. (7) seems not a scalar?
8. Line 213, what does the loss function mean? Why do you design it? Also, is the state function also trained by it?

# Significance
1. The improvements are not significant.
2. Environment is a bit narrow. Why only in 3D cases?
3. The authors only study one dataset, making the result not strong enough.
4. Limited ablation studies. No ablation about the loss, the new task encoding scheme, or the new SF approximation. The showed two ablations, in my opinion, are about minor issues.


**Questions:**

I have some puzzles about the paper, plz help me clarify them.

1. Eq. (5), what is the SF dimension? Is $B$ a random variable and what does it stand for? How $b_k$ take values?

2. I don't understand why you can write SF $\psi_{\theta}^k$ as the Expectation of some distribution. Do you mean that $b_i$ should be $\gamma^i$ and $\mathrm{Exp}(l_{\theta}(s_t,w,e_k))$ should be $\phi_{t+1+I}$?

3. Is there any difference between saying $\psi_{\theta}^k=B^T l_{\theta}^{\psi}$ is an Linear combination  of all $\mathrm{Exp}(l_{\theta}(s_t,w,e_k))$ and saying it is an expectation of $p_{\psi^k}? Why do you prefer the latter?

4. What does Line 144 mean? What is the difference between $\tilde{\psi}^{\pi_k},\psi^{\pi_k},\tilde{\Psi}^{w}$? I look into the "Universal Successor Feature Approximator" paper. It seems the tilde stands for approximation. What confuses me is why it only approximates the neural network $\psi_{\theta}$? Shouldn't they be the same thing?

5. Do you lose a $t+1$ in $\phi_{\theta}(s_t,a_t)$ in Eq. (4)?



**Limitations:**

Limited scenarios. The computation burden seems to be increased compared with previous methods. The networks can be harder to train due to complicated designs. Computing approximations from sampling of distributions can also heavy the burden of computation both in training and inference.

---

> ### Author Rebuttal · Authors · 2023-08-09
>
> Thank you for taking the time to read this paper and for asking these clarifying questions. We have used your comments to improve the text. Due to space constraints, we respond to the most pressing comments and questions below.
>
> # Environment is a bit narrow. Why only 3D case? Using only one dataset makes the result not strong enough.
>
> We emphasize that Playroom is a significant increase in complexity over prior SF work and discuss this in more detail in the shared response. We also significantly improve over the standard Playroom baseline, IMPALA. We acknowledge that applying our work to other domains would be interesting. We have added this discussion to the main text.
>
> # The improvements are not significant
>
> Can you please provide specific examples for why this is not significant? In 5.1, baseline methods fail at our tasks. In 5.2, we learn with 100 million+ fewer samples. Additionally, this is the first demonstration of transfer in a 3D environment with SF&GPI where all necessary representations are discovered.
>
> # Is computation burden increased?
>
> Please see the joint response.
>
> # Clarifying how SFs are computed
>
> First note that we can represent any scalar $y$ as a weighted average of some of fixed bins $\mathbb{B}=\\{b_1, \ldots, b_M\\}$ and learnable weights $\mathbf{p} = \\{p_1, \ldots, p_M \\}$ as $ y \approx \sum^M_{m=1} p_m b_m$ where $b_m \in \mathbb{R}$ and $p_m \in \mathbb{R}^+$. $(\mathbb{B},\mathbf{p})$ define a pmf. We can learn $\mathbf{p} = \operatorname{softmax}(f_{\theta}(x)) \in \mathbb{R}^M$ for inputs $x$ with a negative log-likelihood objective [1].
>
> Now, consider approximating an n-dimensional SF-vector $\psi^{\pi_w}(s,a)\in \mathbb{R}^n$ with $\psi_{\theta}(s,a, w)$ using a pmf. Assuming all SF values can be represented with bins $\mathbb{B}$, we can compute $\psi_{\theta}(s,a, w)$ as
>
> $$
> \psi_{\theta}(s,a, w) = \\{\psi^k\_{\theta}(s,a, w)\\}^n_{k=1}
> \quad
> \psi^k\_{\theta}(s,a, w) = \sum^M\_{m=1} p^{\psi\_k}\_m b\_m
> \quad
> \mathbf{p}^{\psi\_k} = \operatorname{softmax}(l\_{\theta}(\ldots))
> $$
>
> where $\mathbf{p}^{\psi_k}$ are the weights for the $k$-th SF pmf. Leveraging this strategy for learning value-functions is known to improve learning by stabilizing gradients [1]. We highlight that SFs are types of value functions.
>
> [1] Improving Regression Performance with Distributional Losses
>
> # Eq 5, what is the SF dimension?
>
> $\psi_{\theta}(s,a, w) \in \mathbb{R}^n$ and $\psi^k_{\theta}(s,a, w) \in \mathbb{R}$.
>
> # What is B?
>
> We defined $\mathbb{B}=\\{b_1, \ldots, b_M\\}$ with $M$ real numbers evenly spaced between $b_1$ and $b_M$.
>
> # What does it mean to write SFs as the expectation of some distribution?
>
> It's the same as $\psi^k_{\theta}(s,a, w) = \sum^M_{m=1} p^{\psi_k}_m b_m$. We have changed the text to use this instead.
>
> # Righthand side of Eq 7 seems not a scalar?
>
> Both $\operatorname{twohot}\left({y_t^{\psi_k}}\right)$ and $p^{\psi_k}$ are length-$M$ vectors, so the righthand side of eq 7 is a scalar. We have clarified this in the text.
>
> # What does $\theta^o$ mean in Eq 6 & 7?
>
> During Q-learning, we have a target $y_t = r_{t+1} + \gamma Q_{\theta}(s_{t+1}, a^*)$ that is *non-stationary* because $\theta$  is changing. We mitigate this with *target parameters* $\theta^o$ that update at a lower frequency. We have clarified this in the text.
>
>
>
> # Why do we omit ${L}_{\psi}$? Are we aligning SFs and cumulants?
>
> The original learning objective for USFA/MSFA has $L_\psi$, $L_r$, and $L_Q$. Here, $L_\psi$ aligns $\psi_{\theta}(s,a, w)$ with $\mathbb{E}\_{\pi\_w}\left[\sum^{\infty}\_{i=0} \gamma^i \phi_{t+i+1}\right]$ via a regression loss. We **replace** $L_\psi$ with $L^{\tt cat}_{\psi}$, a negative log-likelihood loss. Both estimate the same expectation.
>
> # Line 213, what does the loss function mean? Does this train the state function?
>
> This loss indicates that we are performing gradient ascent with gradients $A_t \nabla \log p_{\theta}\left(\alpha \mid s_t, w\right)$, where $A_t = R_t - V_{\theta}\left(s_t\right)$ is the "advantage" of the coefficient chosen $\alpha$ at time $t$. Here, $R_t = \sum_{i=0}^{\infty} r_{t+i+1}$  is the experienced return and $V_{\theta}\left(s_t\right)$ is the predicted return. Optimizing this increases the likelihood of choosing coefficients in proportion to the "advantage" $A_t$.
>
> Yes, this trains the new state function. We have changed the text to clarify this.
>
> # Only 2 ablations
>
> Could you please point out what ablations you believe are missing? We provided the following:
>
> Figure 4:
>
> 1. no categorical: replaces pmf representation and negative-log-likelihood loss with scalar representation and regression loss.
> 2. no stop gradient: ablates stopping gradients in loss.
> 3. no $||w||$: ablates bounding norm of task encoding.
> 4. independent: replaces sharing parameters across SF-dimensions with using different parameters for each dimension.
>
> Figure 7:
>
> 1. Deep OK - binary: replaces bernoulli distribution with gaussian distribution.
> 2. Deep OK - CSFA: replaces CSFA with USFA for estimating SFs during transfer.
>
> Figure 14: number of pmf bins
>
> Figure 15: number of resblocks in cumulant network
>
> # Q-value as linear combination of SFs is somewhat narrow
>
> If SFs capture rich features, we think this dot-product has great potential for generalization. CLIP has demonstrated that a dot-product between visual features and language encodings can provide powerful generalization. This is analogous to our dot product between SFs (visual features) and task encodings (language encodings).
>
> # networks can be harder to train due to complicated designs.
>
> Can you please specify what part of our method might complicate training?
>
> # What does Line 144 mean?
>
> We were trying to say that one can parameterize an SF-approximator for a policy $\pi_w$ with an encoding of the task that defines it, i.e. we can approximate $\psi^{\pi_w}$ with $\psi_\theta(s, a, w)$. We have simplified the text.
>
> # Do you lose a $t+1$ in eq. 4?
> Yes we did.

---

> > ### Comment · Area_Chair_gD7P · 2023-08-16
> > **Reviewer z7ra**
> >
> > Dear Reviewer z7ra,
> >
> > Could you please read the author's rebuttal and other reviews, and indicate whether it has changed your opinion?  You can also engage further with the authors.  Thank you.
> >
> > Best, AC

---

### Author Rebuttal · Authors · 2023-08-09

We thank the reviewers for their very constructive feedback which we have used to improve the clarity and accessibility of our paper. We are encouraged that reviewers found that we tackle a novel challenge [R2, R3] with a well-reasoned approach [R3], have significant [R4] and convincing results [R2, R3] on a challenging problem [R2, R3, R4], with figures that help in understanding the method [R1, R2, R3] and writing that is clear and easy to understand [R2, R3]. Note that we have labled reviewers as follows: (R1: z7ra, R2: RZN8, R3: xth2, R4: cEsx).

We respond to individual comments below but provide some common responses here. We have also added a figure to explain how we compute a successor feature with a learned probability mass function.

**Does CSFA use more compute or have more complex calculations compared to USFA or MSFA?**

We clarify that the calculations of CSFA are only slightly more complex than USFA/MSFA. Prior work that has learned Q-values, SFs, and cumulants has had 3 terms, $L=L\_\psi + L\_r + L\_Q$ [1-3] (we omit coefficients for simplicity). In our new objective, we **replace** $L\_\psi$ with $L^{\tt cat}\_{\psi}$. We accidentally omitted $L\_Q$ when describing the losses used by prior work in section 3. $\mathcal{L}^{\tt cat}\_\psi$ differs in that it estimates SFs with a negative log-likelihood loss whereas $\mathcal{L}\_\psi$ estimates SFs using mean-squared-error.

We acknowledge that modelling SFs with a pmf does use more parameters, since you need parameters for every bin; however, these computations at the last layer are negligible compared to the rest of the network. We controlled for the number of parameters across SF-based methods by (1) increasing the size of the SF-estimators so that they were approximately equal and (2) keeping all other parts of the architecture the same between methods. CSFA used 7.8M parameters for its SF-estimatr, USFA used 8.1M parameters, and MSFA used 7.5M parameters. Once we do this, the computations in the very last layer where the loss is computed (a softmax followed up a dot-product, eq. 7) is negligible compared to the computations in the rest of the network with millions of parameters. For example, much more computation is done by the ResNets for the agent's vision core and cumulant network. These were shared across all SF-based methods. We have highlighted this in the experiments section of the text.

**Clarifying the contribution from studying playroom relative to prior experiments in the successor feature literature**

Here, we highlight that this work is a significant increase in complexity from prior successor feature literature [4-8]. The most complex task studied by prior work were "go to" tasks in the DMLab environment by [6]. Here, no object-interaction is required. (1) Playroom "place near" tasks add object-interaction, while increasing the task horizon and reward sparsity (2) prior work only combined 4 "go to" tasks while we combine 8 Find and 24 place nears tasks (3) prior work used DMLab with 8 actions [9] while Playroom has 46 actions. We have added this discussion to the main text.

[1] Successor Uncertainties: Exploration and Uncertainty in Temporal Difference Learning

[2] Composing Task Knowledge with Modular Successor Feature Approximators

[3] PsiPhi-Learning: Reinforcement Learning with Demonstrations using Successor Features and Inverse Temporal Difference Learning

[4] Successor Features for Transfer in Reinforcement Learning

[5] Transfer in Deep Reinforcement Learning Using Successor Features and Generalised Policy Improvement

[6] Universal Successor Features Approximators

[7] Composing Task Knowledge with Modular Successor Feature Approximators

[8] The Option Keyboard Combining Skills in Reinforcement Learning

[9] DeepMind Lab

---

### Decision · Program_Chairs · 2023-09-21

**Decision:**

Accept (poster)

**Comment:**

This paper presents a method for transferring behavioral knowledge across tasks by jointly discovering state-features and task encodings while estimating successor features. The paper received 1 strong accept, 2 borderline accept, and 1 borderline reject recommendations from reviewers. The reviewers appreciated the challenging problem of jointly learning task encodings, cumulants and successor features, sound approach, thorough experiments, and good results. Questions and concerns included lack of discussions on some approach components, some related work (such as hierarchical, hindsight, and meta-learning approaches), compute/complexity, and Deep Option Keyboard optimization. Most of these concerns were addressed in the rebuttal and discussion. Some remaining concerns included lack of empirical evidence on how to improve/build on Deep Option Keyboard to obtain optimal performance. The borderline reject reviewer did not follow up on the post-rebuttal discussions, and also indicated their low confidence in their review – the ACs felt that this reviewer’s major questions and concerns were adequately addressed by the rebuttal. Overall, after carefully considering the paper, rebuttal, and discussions, the ACs feel that the positives outweigh the negatives, and recommend acceptance. It is recommended that the revision include a discussion on computation complexity and task limitations.